# The conserved microRNA miR-34 regulates synaptogenesis via coordination of distinct mechanisms in presynaptic and postsynaptic cells

Elizabeth M. McNeill [1,4], Chloe Warinner[1], Stephen Alkins[2], Alicia Taylor[1,4], Hansine Heggeness[1], Todd F. DeLuca [1], Tudor A. Fulga [1,5], Dennis P. Wall [3], Leslie C. Griffith[2] & David Van Vactor [1✉]

Micro(mi)RNA-based post-transcriptional regulatory mechanisms have been broadly implicated in the assembly and modulation of synaptic connections required to shape neural circuits, however, relatively few specific miRNAs have been identified that control synapse formation. Using a conditional transgenic toolkit for competitive inhibition of miRNA function in *Drosophila*, we performed an unbiased screen for novel regulators of synapse morphogenesis at the larval neuromuscular junction (NMJ). From a set of ten new validated regulators of NMJ growth, we discovered that *miR-34* mutants display synaptic phenotypes and cell type-specific functions suggesting distinct downstream mechanisms in the presynaptic and postsynaptic compartments. A search for conserved downstream targets for miR-34 identified the junctional receptor CNTNAP4/Neurexin-IV (Nrx-IV) and the membrane cytoskeletal effector Adducin/Hu-li tai shao (Hts) as proteins whose synaptic expression is restricted by miR-34. Manipulation of miR-34, Nrx-IV or Hts-M function in motor neurons or muscle supports a model where presynaptic miR-34 inhibits Nrx-IV to influence active zone formation, whereas, postsynaptic miR-34 inhibits Hts to regulate the initiation of bouton formation from presynaptic terminals.

[1] Department of Cell Biology and Program in Neuroscience, Harvard Medical School, Boston, MA 02115, USA. [2] Department of Biology and Volen National Center for Complex Systems, Brandeis University, Waltham, MA 02454, USA. [3] Department of Pediatrics, Division of Systems Medicine, Stanford University, Palo Alto, CA 94305, USA. [4] Present address: Department of Food Science and Human Nutrition, Iowa State University, Ames, IA, USA. [5] Present address: Weatherall Institute, Oxford University, Oxford, UK. ✉email: davie_vanvactor@hms.harvard.edu

The molecular mechanisms controlling synapse formation, maturation, and stability have been studied in many contexts revealing extensive networks of signaling and structural proteins required on both sides of this specialized cellular junction. Understanding the regulatory logic that deploys and tunes these proteins to coordinate synapse morphogenesis simultaneously in both presynaptic and postsynaptic compartments remains an important research frontier.

Micro(mi)RNAs have emerged as ideal candidates to regulate nervous system development, plasticity and function[1,2]. miRNAs can concurrently regulate the stability and translation of distinct target mRNAs via complementary miRNA response elements (MREs) that rely on conserved "seed" sequences to confer robust yet imprecise base-pairing with target transcripts[3,4]. After initial discovery of these small non-coding RNAs and key downstream loci by genetic analysis[5–7], genomic and informatics technologies were developed to identify many hundreds of conserved miRNAs and predict their direct target mRNAs in many animal species[8,9]. Subsequent miRNA profiling of neurons and synapses revealed that neurons express a particularly rich landscape of potential regulatory functions with significant potential for cell-type-specific mechanisms[1,10–14].

Driven largely by candidate approaches, a gradually increasing number of individual miRNAs have been found to shape and modulate neural networks. miRNA regulation of individual gene targets is known to control different aspects of synaptic morphogenesis, including dendritic spine modification, axonal sprouting and synaptogenesis [reviewed in Refs. [15–17]]. Even though the number of well-studied miRNAs remains relatively small, it is already clear that these regulators can also exhibit striking temporal and tissue selectivity in their function that can be particularly important within complex multicellular assemblies[18–23]. However, it has been difficult to broadly map the functions of miRNAs within the intact nervous system in order to determine the extent of miRNA regulation and the precise contributions of distinct target genes to synaptic development.

One important resource for comprehensive analysis of miRNA mechanisms in neural development has been the painstaking construction of deletion collections to provide null mutations covering the majority of conserved miRNAs in model organisms[24–26]. Yet, while vital for thorough genetic characterization, null alleles produce chronic and systemic loss of function (LOF), making the advent of conditional tools a priority for dissecting miRNA function in vivo [reviewed by Refs. [27,28]]. For this reason, we developed a collection of transgenic competitive inhibitors ("miRNA-SPonge" [miR-SP]) using the upstream activating sequence [UAS] control of the heterologous transcription factor GAL4 to provide spatio-temporal miRNA LOF in Drosophila[18,29]; this resource was made in a uniform genetic background ideal for quantitative phenotypic comparisons. Analysis of muscle form and function with the miR-SP collection revealed that endogenous levels of tissue-specific miRNA expression do not correlate well with functional impact[29], thus stressing the value of unbiased genetic screening to uncover novel functions.

Here we utilize the miR-SP resource to survey and dissect miRNA control of synapse morphogenesis at the larval NMJ, a popular model glutamatergic synapse where quantitative analysis of multiple synaptic features can be combined with an extensive database of highly conserved effector gene functions to facilitate rapid analysis of specific miRNA-dependent mechanisms [reviewed by Refs. [30,31]]. Using this genetic approach, we discovered many new conserved synapse-regulatory miRNAs, and demonstrated that a single miRNA can independently control distinct synaptic features through tuning expression of distinct targets in the presynaptic and postsynaptic cell.

## Results

**Surveying miRNA regulation of synaptogenesis.** To identify novel miRNA requirements on either or both sides of the synapse, we performed a primary screen using a ubiquitous *tubulin-GAL4* driver combined with each of 131 *miR-SPs* complementary to high-confidence miRNAs not previously known to display NMJ phenotypes (Supplementary Data 1). We quantified type Ib and Is bouton number for the well-characterized NMJ at the cleft between muscles 6 and 7 (M6/7NMJ; Fig. 1a) in wandering 3rd instar larvae (L3) using immunocytochemistry combining a presynaptic membrane marker (anti-Horseradish Peroxidase; HRP) and a postsynaptic marker (the MAGUK scaffolding protein Discs-large; Dlg) with a sampling depth corresponding to over 1200 boutons (10 A2/A3 hemisegments) for control animals that would be sufficient to detect robust phenotypes above 19% change at a statistical power of 0.8 for $p \leq 0.05$. The morphometry values for *miR-SPs* were compared statistically to a *tubulin-GAL4;ScrambleSP* control cross that displayed normal morphology indistinguishable from the *attP2; attP40* genetic background of the collection[29]. In contrast to the relatively small number of gross developmental and lethal phenotypes previously revealed by ubiquitous expression[29], we found a surprisingly large set of *miR-SP* lines (21/131) that induced significant divergence from control bouton number (Supplementary Data 1).

Like other genetic screening methods such as RNA interference (RNAi), *miR-SPs* display a measurable false-discovery rate, either due to off-target effects or limitations in the timing or degree of miRNA inhibition[29]. For this reason, we then performed a secondary screen of available deletion mutations corresponding to our novel candidates in order to provide validation with independent genetic reagents[26]. We found that ten of our novel *miR-SP* NMJ phenotypes match those of null alleles, including *miR-13a, miR-14, miR-34, miR-92a, miR-92b, miR-219, miR-277, miR-316, miR-973*, and *miR-1014* (Fig. 1b compares mean *miR-SP* values in orange and null in blue for each confirmed hit normalized to their respective control strain). The majority of these conserved miRNAs (8/10) were required to promote NMJ growth (e.g. *miR-14* in Fig. 1c), while only two are required to limit NMJ size (e.g. *miR-316* in Fig. 1d). A summary of the screen (Fig. 1e) shows that five of the primary hits were false positives, thus emphasizing the importance of combining analysis of *miR-SPs* with null mutations. Given the fact that six nulls were either unavailable or lethal prior to the 3rd instar, the ten validated hits were likely to represent an underestimate of novel miRNA regulators for NMJ morphogenesis.

Our unbiased screen revealed an unexpectedly large set of miRNAs required for NMJ development, implying a complex network of underlying effector genes. This led us to ask if the hits in our screen could have been anticipated based on the functional nature of genes predicted to be downstream by in silico analysis. With the many available databases for both predicted miRNA targets [TargetscanFly and MicroCosm Targets[9,32–35]] and a large number of functionally annotated synaptic genes in humans or other vertebrates [SynaptomeDB[36]] that are conserved in *Drosophila* [orthologs predicted with Diopt[37]], we plotted a ranking of miRNAs based on the percentage of their predicted target gene pool that represent known synaptic functions (Supplementary Fig. 1a). When we examined the location of our confirmed NMJ hits within this ranking, we found them to be biased to the upper ranking but still widely distributed (red bars in Supplementary Fig. 1a), highlighting the fact that many novel miRNA functions would have been difficult to identify without an unbiased genetic approach.

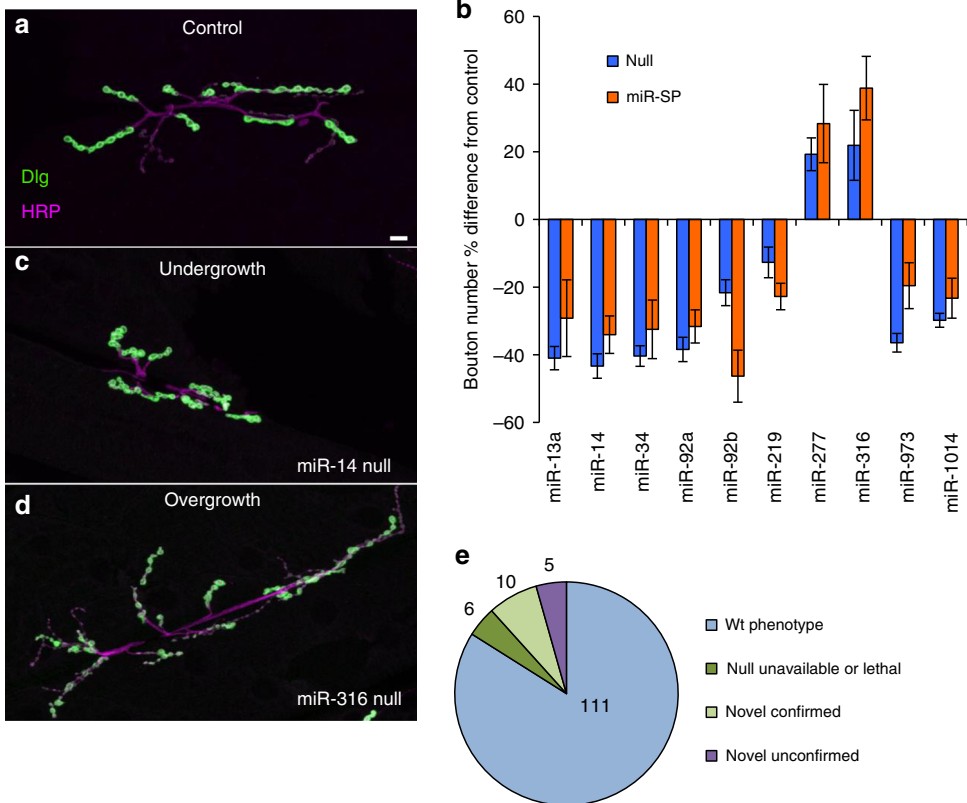

**Fig. 1 In vivo screen of miRNA regulatory complexity in the *Drosophila* NMJ. a**, **c**, **d** Scanning confocal images of NMJ arbors at larval muscles 6 and 7 (L3 segment A2) are shown stained for the presynaptic membrane (HRP, magenta), and postsynaptic cytomatrix (Dlg, green); scale bar (10 μm) applies to all panels. Representative phenotypes from the screen are shown, including **a** $w^{1118}$ control, **c** *miR-14* null undergrowth, and **d** *miR-316* null overgrowth. **b** Bouton number phenotypes that reached or exceeded significance ($p \leq 0.05$) are shown as a percentage change (Error bars indicate + SEM) relative to matched control for GenII *miR-SP* driven ubiquitously using a *tubulin-GAL4* driver (orange bars; *tubulin-GAL4,Scramble-SP* is the control genotype at 122.4 + 11 SEM; sample depth of $n = 10$ hemisegments) or as confirmed by subsequent analysis of *miRNA* null alleles (blue bars; $w^{1118}$ is the control genotype at 123.7 + 8.75 SEM; sample depth of $n = 10$ hemisegments, except in the cases of miR-219 and miR-277 that were sampled and $n = 20$). All results shown were analyzed by ANOVA. Two tailed *T*-test and show *p*-values ≤ 0.05 (see Supplementary Data 1 for screen data). **e** Summary of bouton number defects observed in the *miR-SP* ubiquitous competitive inhibition assay, relative to novel phenotypes confirmed with null alleles (see key for categories). Source data are provided as a Source Data file.

**miR-34 is required for multiple aspects of NMJ synapse Morphogenesis**. One tremendous advantage of the *Drosophila* NMJ for studies of RNA regulatory mechanisms is the wealth of genetic data that define effector genes and pathways important for synapse development in this system[38–42]. To help prioritize miRNAs for deeper characterization, we compiled a database of genes with published larval NMJ phenotypes (Supplementary Data 2) and combined this with our list of *Drosophila melanogaster* (dme) genes orthologous to homo sapiens (hsa) genes with synaptic annotations. In parallel, we took our target predictions, and asked which of these fall within the combined synaptic set. We then ranked our ten novel miRNA candidates with these data as an estimate of enrichment for predicted synaptic target genes (Supplementary Fig. 1b). Several of our miRNA hits ranked highly, although some of these were members of large miRNA families (e.g. miR-92a/b or miR13a). The top-ranked candidate (miR-34) is a highly conserved, single gene miRNA (Supplementary Fig. 1b), whose mature form (dme-miR-34-5p) displays 87% sequence conservation to its three human homologues hsa-miR-34a-c-5p (Supplementary Fig. 1c). With this level of conservation in both the miRNA and possible synaptic target genes, the robust phenotype of miR-34 was an obvious choice for deeper characterization. In addition, although miR-34 family miRNAs had been implicated in different aspects of neural plasticity and development[43–47], the

precise functions of miR-34-family genes in presynaptic development had not been described.

Our analysis of gross L3 M6/7NMJ architecture in *miR-34* LOF animals showed that this miRNA is required to promote motor axon terminal arbor expansion and the formation of presynaptic boutons (Fig.s 1b and 2a-b); however, branching of the NMJ arbor was not dependent on miR-34 (Supplementary Fig. 2g). Moreover, observed decreases of type Ib bouton number at M12 NMJ suggested that miR-34 is generally required for morphogenesis of this class of presynaptic varicosity (Supplementary Fig. 2h). More detailed analysis of type I boutons with specific markers for components that make up key presynaptic structures, such as the T-bars of active zones (AZs) visualized with anti-Bruchpilot (Brp), or the postsynaptic cytomatrix within the subsynaptic reticulum (SSR) visualized with Dlg, demonstrated defects in *miR-34* mutants indicative of altered organization on both sides of the synapse (Fig. 2c, d). Significant increases in Brp intensity (Fig. 2c; Supplementary Fig. 2a–f) and Brp puncta number (Supplementary Fig. 2i) suggested elevated density or assembly of sites for presynaptic glutamate release, consistent with a gross increase in glutamate receptor staining (Supplementary Fig. 2b, e). However, the significant decrease in Dlg intensity (Fig. 2d) suggested a postsynaptic SSR defect. Analysis with an antibody recognizing a second SSR biomarker Syndapin[48] confirmed that miR-34 is required for normal postsynaptic cytomatrix (Fig. 2e), raising the

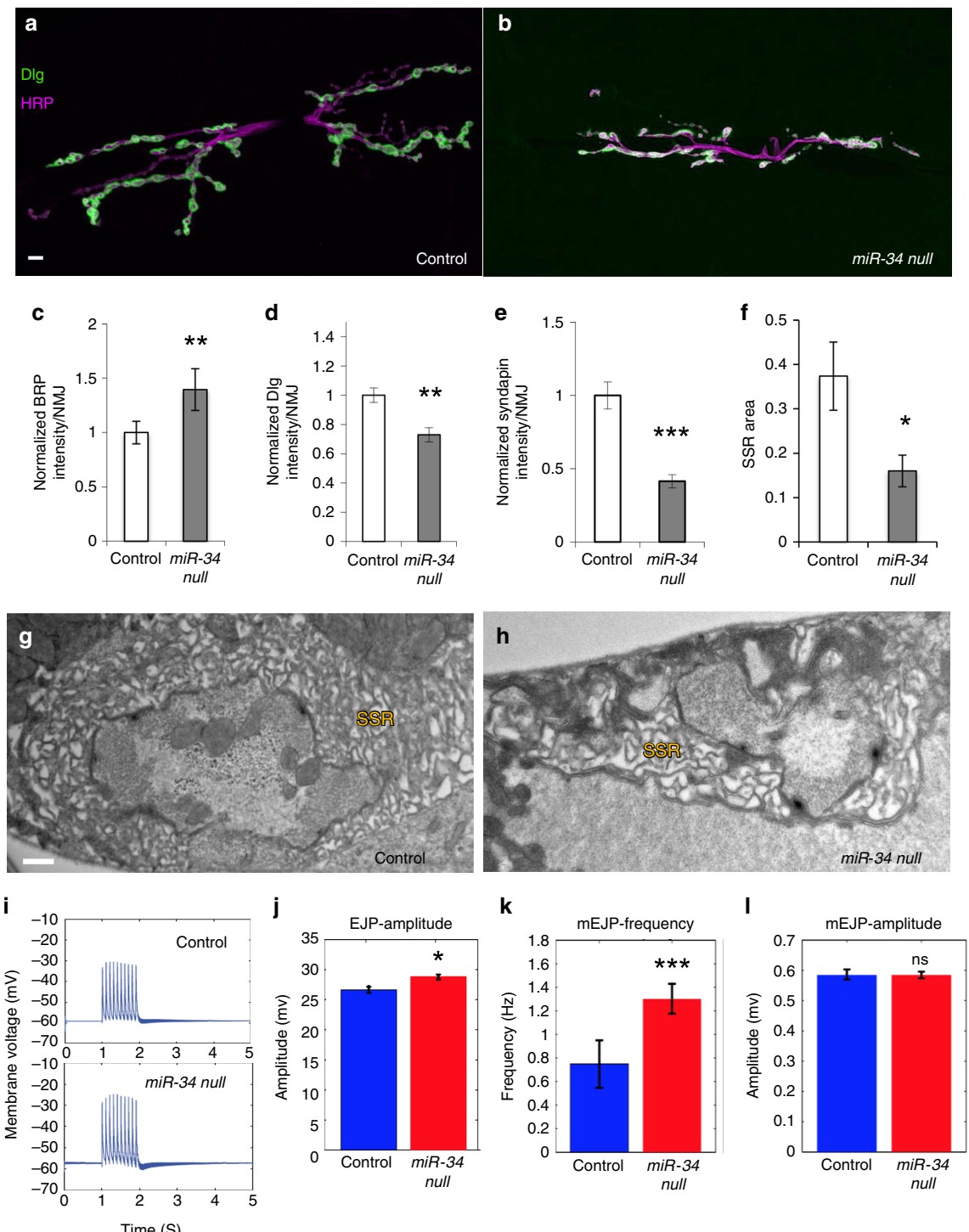

**Fig. 2 miR-34 is required for both presynaptic and postsynaptic development. a, b** Scanning confocal images of the M6/7NMJ (segment A2) were stained for the presynaptic membrane (HRP, magenta), and postsynaptic (Dlg, green). **a** $w^{1118}$ control animal; **b** miR-34 null. Scale bar (10 μm) applies to panels **a** and **b**. **c–e** Intensity quantification of synaptic biomarkers at the M6/7NMJ are shown for control and miR-34 null (** indicates p-value ≤ 0.01; ***p-value ≤ 0.001). **c** Presynaptic T-bar marker Brp imaged using 3-D SIM for high spatial resolution ($n = 6$ control and miR-34 NMJs). **d, e** Postsynaptic marker Dlg ($n = 21$ control and miR-34 NMJs) and Syndapin ($n = 26$ control and $n = 23$ miR-34 NMJs), respectively, imaged with confocal microscopy. **f** Quantification of SSR area from TEM images (* indicates p-value ≤ 0.05) ($n = 12$ boutons from 3 $w^{1118}$ animals; $n = 16$ boutons from 7 miR-34 null animals). **g, h** TEM micrographs of type 1 boutons from the M6/7NMJ; Scale bar in **g**, 500 nm. **g** A $w^{1118}$ control shows presynaptic vesicles are evenly distributed and the postsynaptic SSR is organized into distinct membrane folds. **h** A miR-34 null NMJ is shown for comparison. **i** Electrophysiological recordings from control and miR-34 null M6/7NMJs. Input resistance wasnot significantly different ($p = 0.4178$) between control ($7.9 + 0.4$ MΩ; $n=7$) and miR-34null ($8.2 + 0.6$ MΩ; $n=8$) by Kruskal–Wallis test. **j** miR-34 null animals show slight, butsignificant elevation in evoked EJP amplitude compared to control. Error bars indicate ± SEM, (*p-value < 0.05). (*p-value ≤ 0.05). **k** miniature mEJP frequency was significantly higher in the miR-34 null (***p-value ≤ 0.001), but mEJP amplitude was not significantly (ns) altered in the mutant (**l**). Source data are provided as a Source Data file.

question of whether these alterations correspond to some change in SSR membrane architecture. Moreover, when we quantified a class of boutons that normally lack the SSR in L3 larvae (e.g. type II boutons at M12/13NMJ)[49], we found no significant decrease in the numbers of these small boutons in *miR-34* mutants (Supplementary Fig. 2j), suggesting a correlation between SSR structure and type I bouton addition. To assess postsynaptic structure at high resolution, we next examined *miR-34* null animals at the ultrastructural level using transmission electron microscopy (TEM).

The ultrastructure of normal type 1b synaptic boutons has been extensively characterized[50]. In addition to large mitochondria, the presynaptic nerve terminal contains many clear-core presynaptic vesicles concentrated near electron-dense AZs that include characteristic, central T-bar structures (Fig. 2g and Supplementary Fig. 2m). Type 1b boutons are surrounded by complex muscle membrane folds (SSR; Fig. 2g) that extend glutamate receptor-rich endfeet which may provide signal compartmentalization to sculpt synaptic responses similar to dendritic spine function[51,52]. Analysis of *miR-34* null animals revealed a significant decrease in SSR area compared with control (quantified in Fig. 2f; example in Fig. 2h), consistent with the changes that we observed in Dlg and Syndapin at the light level (Fig. 2d, e). Although synaptic vesicles (SV) were often too dense to count by TEM proximal to *miR-34* null mutant AZs, light level staining for the SV marker Synaptotagmin (Syt) showed normal intensity in the mutant boutons compared to controls (Supplementary Fig. 2k). No significant difference was found in presynaptic area (Supplementary Fig. 2f). However, compared to control (Supplementary Fig. 2m), the AZ ultrastructure of *miR-34* nulls displayed abnormal and diffuse morphology (Supplementary Fig. 2n–p). These observations confirmed that miR-34 is required for both presynaptic and postsynaptic morphogenesis.

The impact of *miR-34* loss on overall bouton number combined with an increase in AZ density and diminished SSR area raised the question of whether synaptic transmission might be altered in this mutant, or whether reduced NMJ arbor was balanced by an increased number of AZs. To answer this question, we recorded and compared evoked excitatory junctional potential (EJP) in control and *miR-34* null larvae (representative EJP traces after 5 V, 10 Hz stimulation are shown in Fig. 2i). We found a relatively small but significant increase in average EJP amplitude (Fig. 2j). Interestingly, a highly significant increase was observed in the frequency of spontaneous release events (mEJPs; Fig. 2k), perhaps consistent with elevation in AZ markers. In contrast, we did not find a significant elevation in mEJP amplitude (Fig. 2l). These data indicate that the mutant NMJs are capable of neurotransmission, and that overall synaptic output (EJP) is surprisingly normal despite significant morphological defects on both sides of the synapse.

**miR-34 plays different roles on the two sides of the synapse.** Although we observed significant structural phenotypes on both sides of the synapse in *miR-34* null animals, it remained possible that these defects reflected a single underlying target gene mechanism acting in either motor neurons or muscle cells. To determine the cellular source of miR-34 action for NMJ growth, we used our *miR-34SP* under control of cell type-specific GAL4 drivers. Interestingly, we discovered that miR-34 is required on both sides of the synapse to achieve normal bouton numbers. Expression of *miR-34SP* specifically in the muscle compartment with *DMef2-GAL4* is sufficient to reduce NMJ growth to levels approaching the severity of ubiquitous expression of *miR-34SP* (Fig. 3b, c, e (red bars); displayed relative to appropriate driver

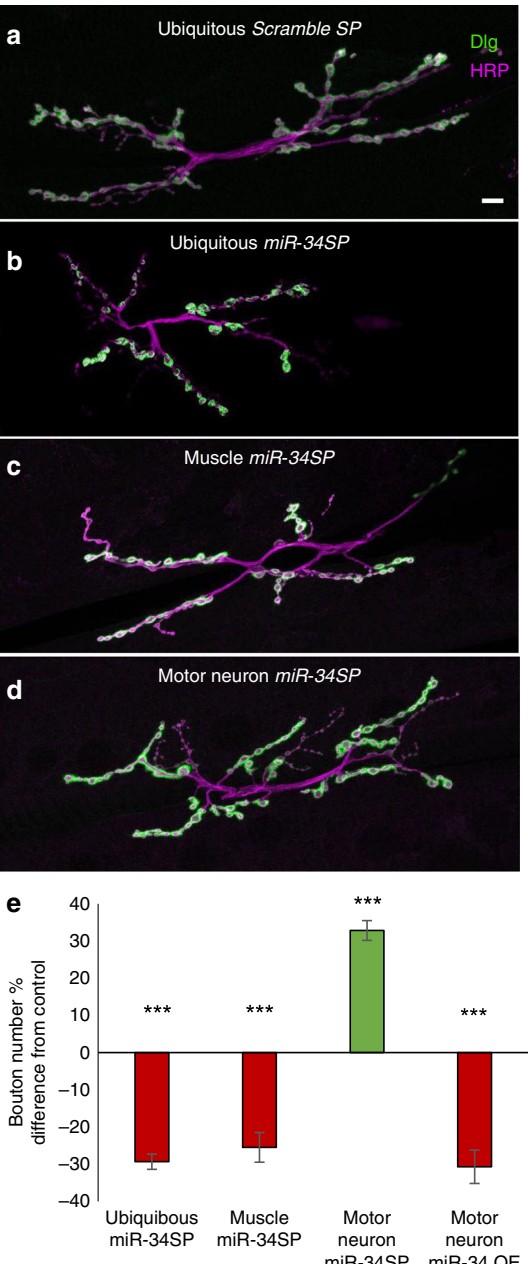

**Fig. 3 Tissue-specific function of miR-34 for bouton addition is revealed by conditional inhibition. a–d** Scanning confocal images of M6/7NMJ (segment A2) stained for the presynaptic membrane (HRP, magenta), and postsynaptic (Dlg, green). **a** ubiquitous *Scramble-SP* expression with *tubulin-GAL4* driver, **b** ubiquitous GenII *miR-34SP* expression with *tubulin-GAL4* driver, **c** muscle-specific GenII *miR-34SP* expression with *DMef2-GAL4* driver, **d** motor neuron-specific GenII *miR-34SP* expression with *OK6-GAL4*. Scale bar (10 mm) applies to **a–d**. **e** Quantification of bouton number changes relative to *Scramble-SP* crossed with appropriate driver control for the GenIII *miR-34SP* competitive inhibitor expressed under control of *tubulin-GAL4* ($n = 20$ for *ScrambleSP* and *miR-SP*), *DMef2-GAL4* ($n = 21$ for *ScrambleSP* and $n = 21$ for *miR-SP*), or *OK6-GAL4* ($n = 20$ for *ScrambleSP* and $n = 21$ for *miR-SP*), or the *UAS-miR-34(+)* overexpressor expressed under control of *OK6-GAL4* ($n = 20$ for OE and control). Significant decrease in bouton number (red bars) or increase in bouton number (green bar) are shown. Error bars indicate ±SEM (***$p$-value ≤ 0.001). Source data are provided as a Source Data file.

crossed with *Scramble-SP* controls) or the null mutation (Fig. 1d). However, when we inhibited miR-34 presynaptically using a motor neuron-specific driver (*OK6-GAL4*) we observed the opposite phenotype resulting in NMJ overgrowth (Fig. 3d) with a significant increase in bouton number (Fig. 3e, green bar). Because miR-34 inhibition in motor neurons gave a surprising result, we asked if overexpression of miR-34 might yield a reciprocal effect. Indeed, expression of a *UAS-miR-34(+)* strain[53] under control of *OK6-Gal4* caused a significant decrease in type 1 bouton number relative to the *OK6-Gal4* control (Fig. 3e, far right red bar); unfortunately, expression of *UAS-miR-34(+)* in muscle was early lethal. These results indicate that miR-34 plays different roles in the presynaptic and postsynaptic compartments, suggesting a working hypothesis that the key downstream target genes in motor neurons and muscle cells are distinct.

**A search for conserved miR-34 targets identifies synaptic effector proteins**. To help elucidate the contrasting synaptic roles of miR-34 in motor neurons and muscles, we set out to identify target genes relevant to NMJ morphogenesis. MRE search algorithms could predict a large number of candidate targets (376), however, because MREs alone are insufficient to predict biological target activity [reviewed in Ref. [54]], we expected many candidates to be irrelevant at the synapse. Thus, we next defined the overlap of potential miR-34 targets with our database of synaptic effector genes including known *Drosophila* NMJ phenotypes (Supplementary Data 2) plus orthologs of genes annotated in SynaptomeDB[36] that could be idenfied using Diopt. This overlap identified 56 candidate synaptic genes. However, MRE sequence prediction alone is not consistently correlated to functional miRNA-dependent translational repression[54]. Moreover, rapid evolution of these short regulatory sequences is thought to produce many species-specific target genes[55]. Therefore, we asked how many of the 56 predicted miR-34 synaptic targets displayed significant sequence conservation between *Drosophila* and human for both the encoded structural gene and the miR-34 seed sequence complement. This final criterion yielded only three conserved proteins with documented synaptic functions that were likely to be regulated by miR-34 from ecdysozoa to vertebrata: Adducin/Hu-li tai shao [Hts][56–59], CNTNAP4/Neurexin-IV[Nrx-IV][60,61] and LRRC7(Densin-180)/Lap1.

Although LRRC7/Densin-180/Lap1 has yet to be studied at the *Drosophila* NMJ, Nrx-IV and Hts are known to control aspects of NMJ development that overlap with the phenotypes we discovered in *miR-34* mutants. In addition to the glial functions of Nrx-IV to promote cell–cell and junctional interactions[60,62,63], this conserved cell surface molecule also regulates the assembly of AZs and bouton addition in presynaptic motor nerve terminals. Indeed, elevation of Nrx-IV is sufficient to significantly increase bouton addition and Brp accumulation[60], thus phenocopying presynaptic effects of miR-34 inhibition. No muscle functions for Nrx-IV have been described; however, the cytoskeletal regulator Hts has been shown to play roles in both motor neurons and muscle cells at the larval NMJ. Presynaptic function of Hts controls exploratory membrane protrusion and synapse elimination, presumably by regulating actin cytoskeleton[56]. In muscle, Hts has been shown to regulate the assembly of the postsynaptic cytomatrix[57,64], including SSR localization of the scaffolding protein Dlg that we find to be dependent on miR-34. These data thus lead us to a working model that the predicted and conserved miR-34 targets Nrx-IV and Hts might account for distinct aspects of the *miR-34* synaptic phenotype.

**Nrx-IV can account for the presynaptic function of miR-34**. Under normal conditions, the junctional adhesion receptor Nrx-IV is highly expressed in glial cells that ensheath motor axons as they enter the target area[65,66] (Supplementary Fig. 3a, b). Interestingly, Nrx-IV protein also accumulates in motor neuron terminals, where it clusters adjacent to presynaptic T-bar structures[60]. Using confocal microscopy and 3-D SIM to image presynaptic boutons beyond the glial footprint, we confirmed Nrx-IV localization in puncta that surround the Brp-positive center of each AZ (Fig. 4a). However, when we compared *miR-34* mutant and control NMJs under identical image acquisition conditions, we discovered a consistent and significant increase in the size and intensity of endogenous Nrx-IV puncta (Fig. 4b; quantitative comparison in Fig. 4c from 3-D SIM data); this was consistent with the presence of a conserved miR-34 MRE in the Nrx-IV 3′ untranslated region (UTR; Fig. 4d). In contrast, Nrx-IV staining intensity along the main axon arbor where glial cells reside did not show a significant increase in *miR-34* mutants (Supplementary Fig. 3c).

To test the prediction that elevation of Nrx-IV selectively in motor neurons might be sufficient to phenocopy the NMJ growth phenotype of *miR-34*, we used the *OK6-GAL4* driver to over-express Nrx-IV (under the control of GAL4 UAS). As predicted, elevation of Nrx-IV in motor neurons induced a significant increase in type 1b bouton number that was qualitatively comparable to motor neuron-specific inhibition of miR-34, albeit quantitatively milder when normalized to the GAL4 control (Fig. 4e). To show that Nrx-IV is required for the NMJ overgrowth induced by presynaptic inhibition of miR-34, we then performed a rescue experiment where Nrx-IV was reduced by RNAi in motor neurons that also express *miR-34SP*. When compared to *scrambleSP* controls driven by the same *OK6-GAL4*, reduction of Nrx-IV effectively rescued presynaptic growth in the *miR-34SP* background (Fig. 4f), thus supporting a model where miR-34 limits presynaptic growth by attenuating expression of Nrx-IV. To be thorough, we also performed a control to determine if a second, inert UAS transgene (*UAS-GFP-L10a*) might simply reduce the efficiency of the *UAS-miR-SP* by diluting *GAL4* activity; however, the phenotype of *miR-34SP* was not significantly altered by the second UAS site (Supplementary Fig. 3d). Interestingly, *OK6-GAL4;UAS-Nrx-IV^{RNAi}* alone caused significant increase in bouton number (Supplementary Fig. 3e). Although regulation of Nrx-IV expression could account for presynaptic *miR-34* phenotypes, it remained formally possible that Nrx-IV might exert control of NMJ growth on both sides of the synapse. Thus, we used muscle-specific *DMef2-GAL4* to drive postsynaptic overexpression of Nrx-IV; however, no significant change in bouton number was observed (Fig. 4e, gray bar), further supporting the model that miR-34-dependent restriction of Nrx-IV expression is required selectively in motor neurons.

**Postsynaptic regulation of hts accounts for miR-34 control of bouton addition**. The *Drosophila Hts* gene produces at least four protein isoforms through differential RNA splicing. Although multiple Hts isoforms are likely to be expressed in L3 body wall muscle[64], the major isoform(s) thought to accumulate in the postsynaptic SSR contain the C-terminal MARCKS domain that binds to F-actin and is recognized the antibody Hts-M[56,57,67] (see Supplementary Fig. 4a for a diagram of the Hts transcription unit and Hts-M antigen). We confirmed this corona-like localization surrounding large boutons using anti-Hts-M in combination with presynaptic HRP (Fig. 5a, a″). However, when we compared Hts-M intensity in *miR-34* mutants versus genetically matched controls using identical confocal imaging conditions (Fig. 5a, a″ versus b, b″), we found a consistent and significant elevation of the Hts-M antigen by at least 1.5-fold within the synaptic

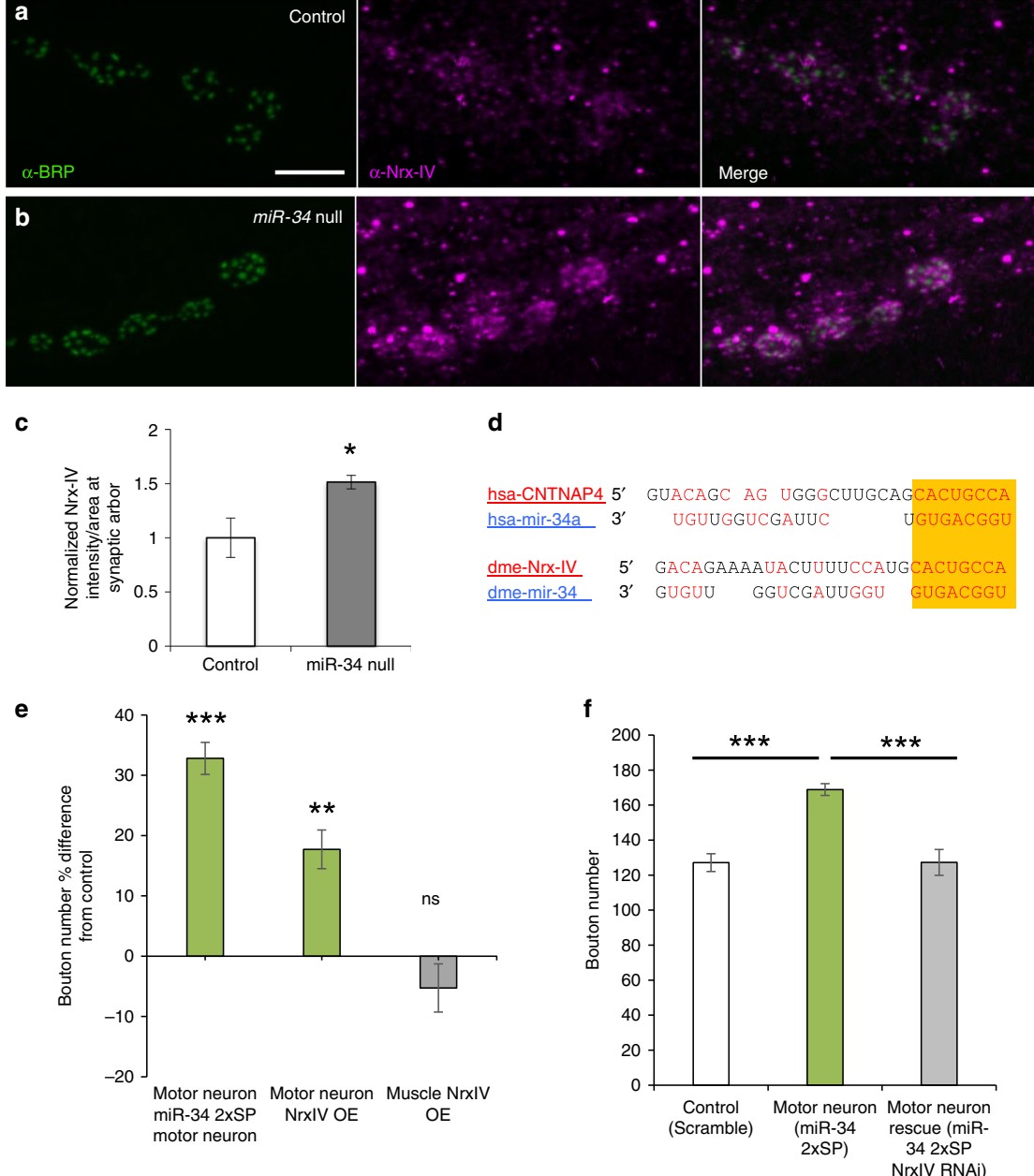

**Fig. 4 Restriction of synaptic Nrx-IV protein can account for motor neuron inhibition of *miR-34*. a, b** Presynaptic active zones marked by the T-bar constituent Brp (green) are surrounded by small punctate accumulations of Nrx-IV protein (magenta) in the terminal boutons of the M6/7NMJ (segment A2; Scale bar: 5 μm). **b** In *miR-34* null animals, an identical staining and imaging protocol reveals a consistent and significant increase in the size and intensity of Nrx-IV puncta compared to control in **a**. **c** The difference in intensity between synaptic Nrx-IV signals is quantified ($n = 6$ NMJs per genotype; \**p*-value ≤ 0.05). **d** Alignment of the conserved MRE for Human (hsa) miR-34a within the 3′UTR of CNTNAP4 is shown in comparison to the corresponding MRE for *Drosophila* (dme) miR-34 in the Neurexin-IV (Nrx-IV) mRNA. **e** Motor neuron-specific competitive inhibition of miR-34 ($n = 21$) using *OK6-GAL4* to express *miR-34SP* is compared to Nrx-IV overexpression (Nrx-IV OE) of under *OK6-GAL4* (green bar; $n = 19$) or a muscle-specific *DMef2-GAL4* (gray bar; $n = 20$) (\*\**p*-value ≤ 0.01; \*\*\**p*-value ≤ 0.001) all relative controls $n = 20$. **f** *OK6-GAL4* expression of *miR-34SP* induces a significant increase in type 1 bouton numbers per NMJ (green bar; $n = 20$) compared to *Scramble-SP* controls (white bar; $n = 21$); this phenotype is completely rescued by co-expression of *UAS-Nrx-IV*[RNAi] (gray bar; $n = 20$) (\*\*\**p*-value ≤ 0.001). Source data are provided as a Source Data file.

compartment (quantified in Fig. 5c). There was also a marked increase in Hts-M expression throughout the muscle fibers (compare Fig. 5a, b, and a″ to b″). These data suggested that synaptic Hts-M isoform levels are negatively regulated by miR-34, consistent with the presence of a conserved miR-34 MRE in the 3′ UTR downstream of the MARCKS domain-encoding exon (Fig. 5d; Supplementary Fig. 4a).

To test whether selective elevation of the Hts-M isoform(s) might be sufficient to mimic the NMJ defects that we observed in either *miR-34* null mutants or *miR-34SP* competitive inhibition, we used a wild type *UAS-hts-M*[wt] cDNA transgene lacking the miR-34 3′ MRE[56] for cell type-specific overexpression of Hts-M. Using a *Dmef2-GAL4* driver, we found that a single transgene insertion (*UAS-hts-M*[wtVK33]) displayed an intermediate increase

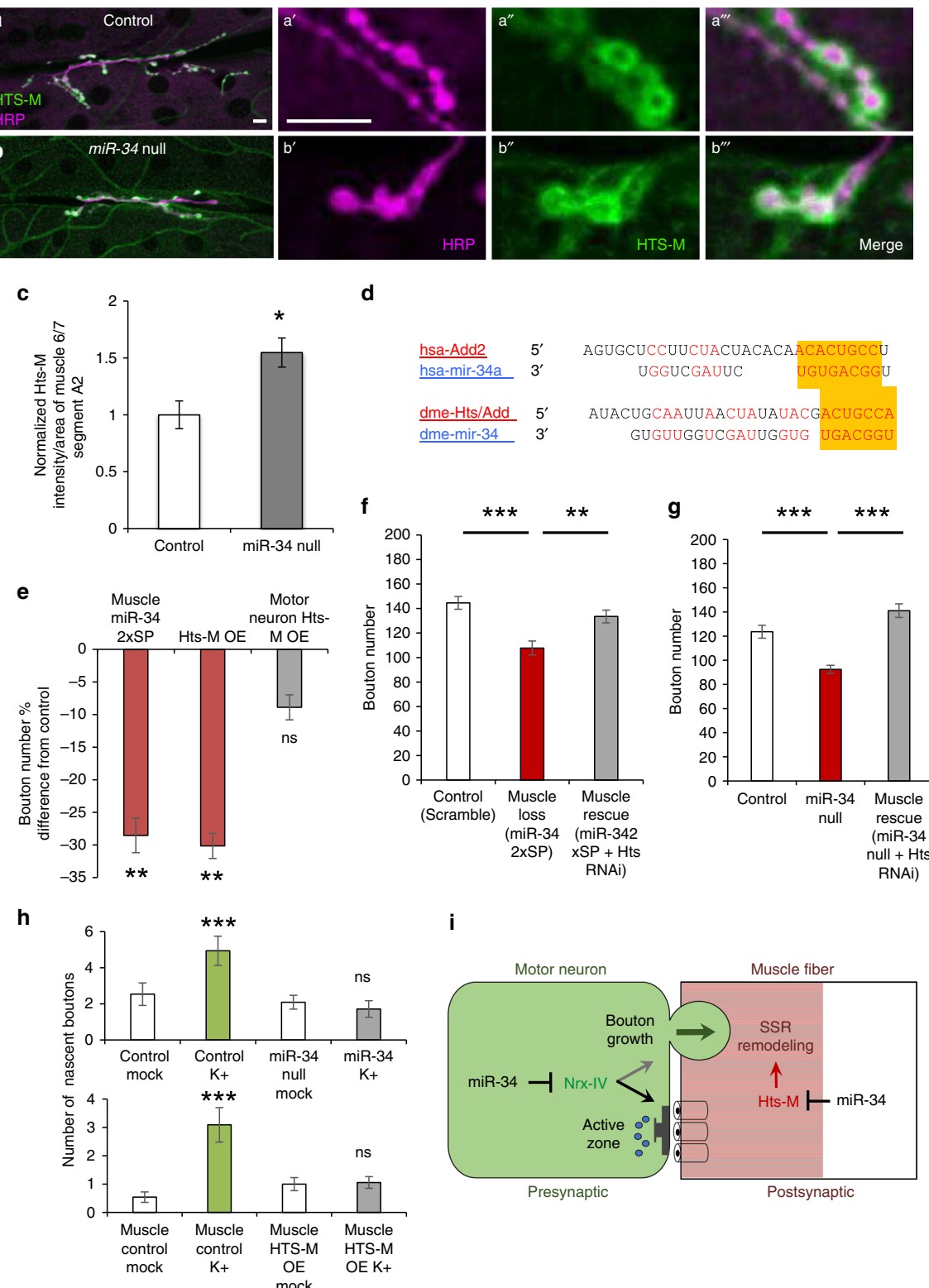

in Hts-M expression compared to control or a 2x UAS-hts-M insert (Supplementary Fig. 4d). When we counted mature type I boutons in the single copy *UAS-hts-M^{wtVK33}* combined with *DMef2-GAL4*, we discovered that bouton number decreased at the muscle 6/7 NMJ by nearly 30%, producing a phenotype that was indistinguishable from that observed in *DMef2-GAL4;miR-34SP* (Fig. 5e); this phenotype was also highly reminiscent of the effect on the muscle 4 NMJ when *UAS-hts-M* was expressed with

*DMef2-GAL4* in a previous study[56]. To further verify our finding that Hts-M overexpression in muscle is sufficient to restrict type I bouton addition, we used *DMef2-GAL4* to drive expression of a single insertion cDNA transgene from a separate source (*UAS-hts^{S704S}* from Wang and colleagues[64]) in comparison to the matched GAL4 control. We observed a significant albeit milder reduction in type I bouton number (Supplementary Fig.4e). Consistent with previous pan-neuronal overexpression of

**Fig. 5 Regulation of Hts-M isoforms in muscle accounts for miR-34 promotion of bouton addition. a, b** MARCKS domain-containing Hts-M protein isoforms (green) surround presynaptic boutons stained with anti-HRP (magenta) at the M6/7NMJ. Hts-M signal is observed in the muscle with high levels surrounding type 1 boutons in *w1118* control (**a**) and *miR-34* null (**b**) NMJs (segment A2; Scale bar: 10 μm). High magnification insets separating the HRP (**a′, b′**) and Hts-M (**a″, b″**) channels (merged in **a‴, b‴**) reveal increased Hts-M signal in and beyond the SSR halo surrounding large boutons in *miR-34* mutants. **c** Fluorescence intensity profiling of Hts-M antigen within a region of interest surrounding the NMJ and adjacent muscle (see Methods) reveals a significant increase in immunolabeling (* $p \leq 0.05$). **d** Alignment of the conserved MRE for Human (hsa) miR-34a within the 3′UTR of Adducin2 (Add2) is shown in comparison to the corresponding MRE for *Drosophila* (dme) miR-34 in the Hts 3′UTR downstream of the MARCKS domain coding sequence. **e** Muscle-specific competitive inhibition of miR-34 using *DMef2-GAL4* to express *miR-34SP* compared to Hts-M isoform overexpression (Hts-M OE via *UAS-hts-MwtVK33*) using the same *DMef2-GAL4* driver (left and right red bars), reveals highly significant deficits in type 1 boutons that are not significantly different from the decrease observed in *miR-34SP* ($p = 0.79$). However, motor neuron-specific *OK6-GAL4* driven Hts-M OE fails to induce a significant change in bouton number (gray bar); $n = 20$ NMJs for each genotype including control lines which differences are graphed relative to. **f** *DMef2-GAL4* expression of *miR-34SP* induces a significant decrease in type 1 bouton numbers per NMJ (red bar) compared to *Scramble-SP* controls (white bar); this phenotype is completely rescued by co-expression of *UAS-hts*RNAi (gray bar) $n = 20$ NMJs for each genotype. **g** Using *Mhc-GAL4* and the same *UAS-hts*RNAi transgene, we also observe effective rescue of the *miR-34* null type 1 bouton phenotype; $n = 20$ for Control, $n = 19$ for *miR-34* Null, $n = 20$ for Muscle Rescue. **h** Parallel experiments using spaced depolarization to induce activity-dependent nascent boutons show that controls (*w1118* above, *DMef-GAL4* below) show highly significant increases after K+ treatment compared to mock-treated samples (green bars), whereas, both *miR-34* null (above) and UAS-hts-MVK33 over- expression (OE, below) produce no induction (gray bars). For **c** and **e–h**: Error bars indicate ±SEM; **$p$-value $\leq 0.01$; ***$p$-value $\leq 0.01$; ns is $p > 0.05$; For **h** upper panel: *w1118* and *miR-34* mock $n = 13$ for, K+ $n = 9$, *w1118* mock $n = 8$, K+ $n = 11$; For **h** lower panel: *DMef-GAL4* experiments HTS-M OE mock $n = 24$, K+ treated $n = 18$, control mock $n = 13$, K+ treated $n = 11$. **i** A working model of miR-34 action in the presynaptic (green) and postsynaptic compartments of type 1 boutons showing how regulation of Nrx-IV in motor neurons can influence active zones while regulation of Hts-M isoforms controls formation of nascent boutons, possibly via modulation of remodeling of SSR and/or synaptic adhesion. Source data are provided as a Source Data file.

*UAS-hts-M* in a *Hts* mutant background[56], elevation of Hts-M in wild type motor neurons with *OK6-GAL4* did not induce a significant deviation from normal bouton number when compared to matched controls (Fig. 5e, gray bar), indicating that restriction of Hts-M to control bouton number is only required in muscle.

To test whether Hts is necessary for the NMJ growth phenotype induced by postsynaptic miR-34 inhibition, we next used *UAS-hts*RNAi to reduce Hts expression selectively in muscle that was also expressing *miR-34SP* under control of *DMef2-GAL4*. *Hts* knock-down in muscle (*DMef2-GAL4;UAS-hts*RNAi which reduces the Hts-M signal; quantified in Supplementary Fig. 4d) was not sufficiently strong to alter bouton number significantly by itself (Supplementary Fig. 4f). However, when compared to muscle inhibition of miR-34 alone, we found that Hts knock-down effectively rescued the reduction in bouton number caused by inhibition of miR-34 (Fig. 5f). As in motor neurons, addition of *UAS-GFP* had no effect on *miR-34SP* penetrance when driven in muscle (Supplementary Fig. 4g), suggesting that *miR-SP* is not subject to dilution of GAL4. Since *miR-34* null mutants display NMJ growth restriction comparable to muscle-specific *miR-34SP*, we then asked if muscle-specific knock-down of Hts could rescue the null bouton phenotype. Using an independent muscle-specific GAL4 driver (*Mhc-GAL4;UAS-hts*RNAi), we found that Hts knock-down significantly rescued the *miR-34* homozygous null (Fig. 5g). Together, these observations strongly suggest that Hts is both necessary and sufficient to account for the muscle-specific *miR-34* synapse growth phenotype.

Because the restriction in NMJ growth caused by reduction of miR-34 or elevation of the Hts-M cDNA in muscle matched the systemic *miR-34* null phenotype, we concluded that postsynaptic regulation of Hts isoforms is essential for normal presynaptic expansion and thus exerts a dominant effect on NMJ growth. Previous work revealed a role for Hts in preventing bouton retraction that is selective to neurons[56], suggesting that the growth restriction imposed by muscle might act on a different step in bouton formation. Consistent with this idea, we did not observed signs of bouton retraction in *miR-34* null larvae, raising the possibility of a deficit in addition of boutons instead. The first step in bouton formation can be assayed by a convenient spaced KCl depolarization protocol that induces rapid and highly

significant increase in rare nascent "ghost" boutons which can be identified due to lack of maturation markers[68]. Although there was no difference between *miR-34* null and matched control genotypes under mock stimulation, spaced depolarization with KCl revealed that miR-34 is essential for bouton initiation; *miR-34* mutants completely lacked activity-dependent ghost addition (Fig. 5h, top panel). The same assay was then applied to muscle-specific *UAS-Hts-MwtVK33* cDNA overexpression compared to *DMef2-GAL4* alone, revealing a complete deficit in activity-induced bouton formation (Fig. 5h, bottom panel). These data suggest that postsynaptic limitation of Hts-M protein levels by miR-34 regulates the earliest step in bouton formation.

## Discussion
Despite converging lines of evidence implying that many miRNAs contribute to shaping the complex cell-cell interactions that underlie nervous system development, only a small percentage of these post-transcriptional regulators have been examined for such functions in vivo [reviewed by Refs. [1,28,69]]. Consistent with these predictions, our genetic screen of high-confidence miRNAs for synapse morphogenesis phenotypes in *Drosophila* identified a surprisingly large number of novel regulators that could be difficult to predict based on other sources of data. Although many of these new phenotypes satisfied a conservative criterion of genetic validation with null mutations, our deeper analysis of miR-34 highlights the fact that complete systemic LOF can obscure distinct functional contributions of different cell types. Our findings suggest that miR-34 regulates presynaptic features by limiting expression of the junctional receptor CNTNAP/Nrx-IV in motor neurons, whereas miR-34 also limits synapse growth by regulating the F-actin-associated isoforms of Adducin/Hts in postsynaptic muscle cells (Fig. 5i). Null mutations in *miR-34* display combined defects characteristic of Nrx-IV and Hts-M isoform deregulation on the two sides of the synapse. This emphasizes the value of conditional genetic tools to unravel the complexities of miRNA functions in cellular space or time, and raises the question of how often miRNAs control independent processes in neighboring cells.

Unlike miR-34-family target genes such as Syntaxin, Syt, Arc and Sirtuin identified in prior studies of synapse plasticity and

neuronal differentiation in vertebrates[43,45], miR-34 MREs are conserved in the CNTNAP4 and Adducin gene families from insect to human, suggesting that these regulatory mechanisms are ancient. In both vertebrate and invertebrate nervous systems, CNTNAP-family receptors are well known to mediate glial-neuronal and glial-glial cell surface interactions[65,66,70]. However, recent observations at the *Drosophila* NMJ showed that Nrx-IV can modulate AZ assembly and synapse growth[60,63] much like the related receptor Nrx-I[70]. These studies and our findings predict that precise tuning of Nrx-IV levels is essential for normal synaptogenesis, consistent with our finding that miR-34 loss induces comparable presynaptic AZ defects (Fig. 2c and Supplementary Fig. 2a–f, i) as elevation of Nrx-IV[60]. Precise levels of CNTNAP receptors also appear to be vital during human brain development, as dosage changes in CNTNAP4 and CNTNAP2 are associated with autism spectrum disorders[60,71,72]. In addition to altered AZ structure, presynaptic-specific inhibition of miR-34 or elevation of Nrx-IV results in an increased addition of boutons in the motor neuron terminal arbor (Figs. 3e and 4e)[60], suggesting mechanistic coordination between AZ assembly and synapse growth, as has been observed for the peri-AZ scaffolding protein Liprin-alpha and the associated receptor tyrosine phosphatase LAR[70,73]. However, with respect to bouton formation during larval development, the postsynaptic role of miR-34 reveals a muscle imposed limitation on nascent bouton formation, presumably by regulating cytoskeletal structure and cell-cell interaction. This raises the intriguing possibility that mechanisms promoting presynaptic growth must be coordinated with concomitant postsynaptic remodeling, presumably involving one or more trans-synaptic signaling systems. It also remains possible that miR-34 controls additional targets yet to be identified that may contribute to shaping synaptic development, function and/or plasticity. In future, we hope to expand our search for such effectors to better understand the underlying mechanisms and how miR-34 functions in concert with other miRNAs uncovered by our screen.

Previous studies have shown that the MARCKS domain of certain Adducin/Hts protein isoforms mediates a physical bridge between the barbed end of F-actin and the membrane cortex-associated Spectrin-Ankyrin network, thereby conferring stability of cellular structure[56,74–78]. In neurons, this mechanism provides a powerful means for calcium-dependent kinase regulation of the Adducin MARCKS domain to drive presynaptic structural plasticity and formation of memory in response to patterns of neural activity[56,58]; however, relatively little is known about the postsynaptic regulation upstream of Adducin-like Hts-M isoforms. In this regard it is intriguing that miR-34 is required for activity-dependent bouton growth (Fig. 5h), hinting that the miR-34/Hts-M mechanism may be activity-regulated in some way. In addition to a dependence on miR-34 (Fig. 2d–f), SSR architecture has been shown to remodel in response to neuronal activity[79]. Although it is not clear whether SSR at the *Drosophila* NMJ is truly analogous to the dendritic spines of mammalian glutamatergic synapses, as has been suggested[80], mammalian miR-34 family members do control the formation of dendrites and dendritic spines[43]. This raises the possibility that miR-34 regulation of Adducin may play a conserved role in postsynaptic development and/or plasticity.

A tight correlation between bouton addition and SSR structure is also observed in mutants lacking the miRNA miR-8[19]. Analogous to miR-34 regulation of Hts-M, miR-8 promotes bouton addition in larval development by muscle-specific repression of the actin assembly factor Enabled (Ena) that localizes within the SSR[18,19]. Together, analysis of miR-8 and miR-34 suggest that plasticity or dynamics in postsynaptic architecture may be essential for the formation or maturation of new presynaptic varicosities in this system. In addition to cytoskeletal structure,

postsynaptic Hts has been shown to influence the localization of the cytomatrix scaffold Dlg that is required to build the SSR[57]. Interestingly, elevation of all Hts transcripts in muscle[57] has been observed to cause a different effect on NMJ structure than selective elevation of Hts-M (Fig. 5e and Supplementary Fig. 4e), highlighting the question of whether the ratio or local regulation of different isoforms might play an important role in regulating bouton initiation. Because Hts has also been shown to have an impact on additional downstream factors during synapse plasticity, such as CamKII and Par-1[64], the underlying mechanism may also involve a cascade of signaling events. In adult neurons and muscles, miR-34 is expressed and required to protect cells from age-dependent degeneration[29,81–83]. Given Adducin's role in controlling structural stability, it may be interesting to ask if this highly conserved miR-34 target influences the course of neurodegeneration in aging animals.

In conclusion, our analysis of miR-34 revealed novel regulatory relationships required to shape features of the synapse that depend on cellular interaction between presynaptic and postsynaptic partners. The fact that *miR-34* phenotypes reflect combined regulation of distinct target genes in motor neurons and muscle highlights a strength of genetic models where conditional expression or inhibition can be used to illuminate simultaneous contributions of different neighboring cells. Interestingly, like miR-34, the majority of miRNA that regulate NMJ morphogenesis appear to promote bouton addition, including eight of the ten novel miRNA validated with nulls in our study (Fig. 1) and three of four miRNAs previously known to control NMJ growth (miR-124, miR-310-313; miR-8; miR-1000/137)[18,84–86]. This suggests that synapse growth control is tuned by restricting expression of many factors that negatively regulate bouton addition. In the case of Adducin/Hts-M, negative regulation may be achieved by promoting stability or by the need for coordinated cytoskeletal remodeling of the postsynaptic cytomatrix and presynaptic boutons. However, the identity and cellular specificity of target genes downstream of the other novel miRNAs remain to be determined.

## Methods

***Drosophila* strains and genetics**. All stocks were maintained at 25°C according to standard procedures. The *miR-SP* and *Scramble*-SP lines were generated by the Van Vector laboratory; in addition to 131 GenII *miR-SP* lines (*attP-40;attP-2*) used for the primary screen that are now available at the Bloomington Drosophila Stock Center (BDSC, Bloomington, IN, USA)[29], a third generation (Gen III) set of *miR-SPs* was made with identical SP insertion sequences using a pUAST/attB vector with DSRed2[86]; (cloned by Genewiz) and distinct insertion sites (*attP-ZH51C;attP-VK33*; Bestgene strain number 24482 and 9750, respectively) to control for genetic background, and to correct for gradual observed loss of GenII *miR-SP* penetrance over many generations with the 20xSP array in *attP40;attP2* sites since the original primary screens[29]; these new lines are available upon request form the Van Vector laboratory until deposited at the BDSC. Most of the miRNA nulls used in this study were generated by and obtained from the Cohen laboratory[26]. miR-34 null was a kind gift from the Bonini lab[83]. miR-277 null was a kind gift from Chun-Hong Chen. A miR-316 null indel was generated by CRISPR (Supplementary Fig. 5). The UAS-miR34 OE line was obtained from Norbert Perrimon and colleagues. The wild type single insertion Hts cDNA overexpression line, *UAS-hts^wt-attP-VK33*, was obtained from the Pielage lab[56]. *UAS-hts^{S704S}* was obtained from the Kreiger lab[64]. *w1118*, *w6723*, UAS insertion at Nrx-IV *BL17985*, Nrx-IV RNAi *BL28715* and GAL4 lines (*Tub-GAL4*, *OK6-GAL4* and *DMef2-GAL4*), *UAS-GFP-L10a* and *UAS-GFP* were obtained from BDSC. The *UAS-hts^{RNAi}* V29102 was obtained from the Vienna Drosophila Resource Center. For all experiments, we used control genotypes that were as closely matched to the background of the experimental strains as possible; for this reason, we have displayed data in percentage change from control in cases where the genotypes compared required distinct controls that may differ from each other.

**Bioinfomatics**. Ranking of conserved miRNAs was based on percentage of predicted target genes identified in the mammalian SynaptomeDB[36]. For our analysis, the following id mapping was completed using the indicated databases. microRNAs from the NMJ screen were mapped to a current miRBase id and accession number. Accessions were filtered with only mature miRNAs included. Mapping data between mirbase accessions and mirbase ids was completed using miRBase version

19 (ftp://mirbase.org/pub/mirbase/19/aliases.txt.gz). The predicted target genes of the miRs were identified by the following sequence of mappings: miRBase id to a flybase annotation to flybase transcript id to uniprot id to flybase gene id. The following were used for the mappings: microcosm version 5 (http://ftp.ebi.ac.uk/pub/databases/microcosm/v5/arch.v5.txt.drosophila_melanogaster.zip), flybase version FB2013_02 (ftp://ftp.flybase.net/releases/FB2013_02/reporting-xml/FBtr.xml.gz), Uniprot, (version 2013_04, downloaded from the SPARQL endpoint https://sparql.uniprot.org), human SynaptomeDB (version 1.06, http://metamoodics.org/SynaptomeDB/index.php), Uniprot version 2013_04 (mappings downloaded from https://sparql.uniprot.org), and Roundup version 4 (orthologs determined using the following orthology tool: https://github.com/walllab/reciprocal_smallest_distance; for data see Zenodo repository entry: https://doi.org/10.5281/zenodo.3620893).

Ranking of conserved miRNAs was based on percent of target genes which contain a conserved microRNA Regulatory Element (MRE) that have been identified in functional NMJ screens[38,39,41,42]. The gene targets for the human and *Drosophila* miRNAs were predicted from DIANA Tools microT-CDS[87] with a filter stringency set at .5 for both organisms. The target list for *Drosophila* was then submitted to Diopt[37] to identify putative human homologs. This list was then compared to the target gene set identified for humans. This produced a total number of conserved genes with conserved MRE sites for each of the highly conserved NMJ regulators (analysis was not completed for miR-973 and 1014 as they were not represented in all databases at the time of the analysis). This list was compared to the list of identified functional NMJ genes from previous screens and reported as a percentage of all conserved genes with conserved MRE cites.

**Immunohistochemistry and quantification NMJ development**. Wandering third instars raised at a low density were dissected in $Ca^{2+}$-free saline and fixed in fresh 4% paraformaldehyde (vol/vol, Sigma Aldrich) in phosphate-buffered saline for 20 min or for 5 min in Bouin's fixative (Sigma-Aldrich). Larvae were incubated overnight at 4 °C in primary antibodies and for 4 h at room temperature in secondary antibodies. The following primary antibodies were used for immunohistochemistry: Anti-HRP-alexa fluorophore 594 (323-585-021 Jackson ImmunoResearch, West Grove PA, USA), mouse anti-Dlg 4F3, 1:50 and mouse anti-BRP nc82 1:50 obtained from the Developmental Studies Hybridoma Bank Iowa City, IA, USA. Rabbit anti-Nrx-IV (1:1000) (Christian Klambt), Rabbit Hts-M (1:1000) (Lynn Cooley), Rabbit GluRIIB (1:2500) (Aaron Diantonio), Rabbit Syndapin [(1:50), as described,[48]]. Due to a finite supply of the Rabbit anti-Hts-M peptide antibody from the Cooley lab, a new batch of affinity-purified Rabbit antibody was raised to an identical peptide sequence by contract (Primm Lab); this new antibody displays the same SSR localization as the previous serum; moreover this signal is decreased by *UAS-hts*[RNAi] and increased by *UAS-Hts-M cDNA* expression in muscle (this antibody was only used in Supplementary Fig. 4); this antibody is available from the Van Vactor laboratory upon request, while supplies last. Secondary antibodies anti-mouse alexa fluor-488 (Invitrogen), anti-rabbit alexa fluor-568 (Invitrogen) were used at a 1:200 dilution. Larvae were mounted in slowfade gold (Thermofisher scientific) and stored at 4 °C until imaging.

For our primary screen of miR-SP strains, MN 6/7 terminals of muscle 6 and 7 in the abdominal segment A2 and A3 of wandering third instar larvae were used for the quantification of all morphological parameters. This analysis was carried out using a Nikon 90i upright microscope and taking z-stack images using NIS Elements acquisition software and a coolsnap EZ (photometrics) camera. Counting of Type Ib and type Is boutons was then carried out by two independent people blind to genotype counts are reported as a combination of Ib and Is unless otherwise stated. For this primary screen, 10 hemisegments were quantified per genotype (Supplementary Data 1); miR-SP lines that reached $p \leq 0.05$ or greater significance are shown in Fig. 1b (orange bars). For the screen, batches of genotypes ranged from 10 to 12 due to limitations in histology processing, thus a *scrambleSP* control was included in each batch; all samples were distinct. For the secondary validation of phenotypes (Fig. 1b, blue bars), deletion alleles were analyzed at a depth of $n = 10$ hemisegments, and any that failed to reach significance were also analyzed at depth of $n = 20$ in comparison to matched control. For subsequent follow up experiments such as tissue-specific inhibition, overexpression and rescue (Figs. 2–5), our standard sample size was 20 hemisegments. Exact sample sizes are indicated in the figure legends. For the activity-induced bouton initiation assay, a spaced High K+ depolarization paradigm modified pseudo-massed stimulation paradigm was used. Body wall muscles from third instar larvae were dissected in normal-HL3 saline containing 0.2 mM Ca++ and left pinned in a semi-relaxed position. Larvae were subjected to four, 5 min "pulses" in high (90 mM) K+, each separated by 15-min rest in normal HL3. A 15-min rest period after the last pulse was completed and then pelts were stretched and fixed at room temperature in cold 4% PFA for 20 min followed by standard IHC protocol outlined above. Control larvae for all experiments were dissected and incubated using the same protocol but with normal-HL3 alone.

**Confocal and epifluorescence microscopy and image analysis**. Confocal microscopy was performed using a Nikon A1R point scanning confocal with spectral detection and resonant scanner, on an inverted TI microscope and NIS Elements acquisition software. Image stacks of A2 6/7 NMJs were obtained. Prior to acquisition, laser parameters were adjusted to obtain non-saturating conditions, and identical settings were used for control and experimental genotypes. Images were accessed and analyzed using Fiji software and ImageJ 2.0.0-RC68\1.52e.

Fluorescent signal intensity was quantified using ImageJ to determine Raw Integrated Density (RawIntDen = sum of all values in selection for each channel) divided by the area of the region of interest (ROI). To define the presynaptic ROI for quantification of Dlg and Synadapin, a synaptic mask was generated with the ImageJ thresholding tool that captured the presynaptic terminal and the surrounding SSR area by using default settings based on the sum projection of the 3-D volume rendered for the HRP signal, followed by manual selection of the synaptic arbor to generate the ROIs. The ROIs were then applied to the sum projection for the test channel (e.g. Dlg or Syndapin). The final data output of RawIntDen and ROI Area were then generated using the ImageJ Analyze Particles tool. For Hts-M, because a significant amount of Hts-M protein accumulated outside of the HRP-anchored synaptic mask, we used a rectangular ROI selected to capture the entire synaptic arbor, SSR and adjacent muscle field; to avoid bias based on overall NMJ arbor size, we normalized for ROI area. Note that HRP intensity values did not vary significantly between control and miR-34 mutant NMJs. For statistical comparisons, a Student's t-test was employed.

**Structured Illumination super-resolution microscopy**. 3D-SIM data were collected on a DeltaVision OMX V4 Blaze system (GE Healthcare) equipped with a 60×/ 1.42 N.A. Plan Apo oil immersion objective lens (Olympus), 488, 561, and 642 nm diode lasers, and a separate Edge 5.5 sCMOS camera (PCO) for each channel. Z-stacks of were acquired with a z-step of 125 nm and with 15 raw images per plane (five phases, three angles); the axial x–y pixel dimension is 40 nm. Spherical aberration was minimized using immersion oil matching[88]. Super-resolution images were computationally reconstructed to render 3-D volumes from the raw image stacks of Z-planes with a channel-specific measured optical transfer function (OTF) and a Wiener filter constant of 0.001–0.002 using softWoRx 6.1.3 (GE Healthcare). Channel mis-registration was measured using a control slide and multi-channel datasets were registered using the image registration function in softWoRx. Images were accessed and analyzed using Fiji software and ImageJ 2.0.0-RC68\1.52e. For signal intensity quantification of Brp and Nrx-IV, ImageJ tools were used to define regions of interest (ROIs) based on each channel followed by selection of the area surrounding the terminal type 1 boutons of a given NMJ branch. Raw Integrated Density (RawIntDen = sum of all values in selection for each channel) within each ROI was divided by the area of the ROI as generated by the ImageJ Analyze Particles tool. For volume measurements of Brp puncta, we used the 3-D Objects Counter tool in ImageJ on OTF 3-D rendered images after converting images to 16 bit and using the Auto Threshold tool.

**Transmission electron microscopy**. Wandering third instar larvae were dissected as described above and internal organs were removed. Following rapid dissection in Ca+2-free 0.1 M Cacodylate buffer, the body walls were left pinned and fixed at 4 °C overnight in 2.5% paraformaldehyde, 5% glutaraldehyde (vol/vol) and 0.06% picric acid (vol/vol) in 0.1 M Cacodylate, rinsed three times for 20 min in 0.1 M Cacodylate for 2 h on ice and then rinsed three times for 10 min in deionized water and dehydrated in an ethanol series (50, 70, 95, 100, and 100% again, vol/vol) and propylene oxide and placed overnight in 50% TAAB 712 Resin (vol/vol) in propylene oxide. They were transferred to fresh resin for 4 h and then embedded in fresh resin at 65 °C until hard. The 6/7 muscle region was located by eye and the block was trimmed around the desired area. Sections were taken parallel to the surface of the muscles: 50–60 nm sections were collected as a series for a total of 5 um. Sections were mounted on single slot grids, stained 3with lead and uranyl acetate, and imaged on JEOL 1200EX-80 kV electron microscope at 65,000x and 25,000x magnification.

**Electrophysiology**. All larval dissections for electrophysiology were conducted as in Imlach and McCabe[89], using third instar larvae in Hemolymph-Like saline solution 3.1 (HL 3.1) at a pH of 7.1–7.4, containing (in mM) 70 NaCl, 5 KCl, 4 $MgCl_2$, 10 NaHCO$_3$, 5 Trehalose, 115 Sucrose, and 5 HEPES (300 Osm). Body wall muscles were pinned at stretched positions to optimize electrode placement and recording efficacy at neuromuscular junction (NMJ). EJPs were recorded intracellularly from muscle of abdominal segments 3–5 at 20–23 °C in low $Ca^{2+}$ (0.8 mM CaCl$_2$ in HL 3.1 at a pH of 7.1–7.4), immediately following mEPSP recordings. A suction pipette with a tip opening of approximately 10 μm was used to stimulate the segmental nerve (stimulation duration = 0.1 ms, strength = 5 V, rate = 10 Hz). Baseline membrane voltage was recorded for one second prior to stimulation; total stimulation lasted one second; three seconds of responses were recorded. Recordings were discarded if baseline varied by more than 5 mV during stimulation. All EJP recordings were analyzed in MATLAB. Frequency and amplitudes were measured via modified from scripts and functions provided by Ted Brookings. Peter Bronk and Stephen Alkins. Minimum amplitude for mEPSP = 0.4 mV. Post-hoc Jarque-Bera tests were performed to test for normal distribution of data sets. Subsequent Kruskal–Wallis and Two-Sample T-tests of significance were performed between experimental conditions and genotypes ($\alpha = 0.05$). The mean input resistance recorded for control was 7.9 + 0.4 MΩ ($n = 7$) versus 8.2 + 0.6 MΩ ($n = 8$) for *miR-34* null (non-significant difference by Kruskal–Wallis test).

**Reporting summary**. Further information on research design is available in the Nature Research Reporting Summary linked to this article.

## Data availability

The data that support the findings of this study are available from the corresponding author upon reasonable request.

## Code availability

Please see https://doi.org/10.5281/zenodo.3332836 for original code.

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

## Acknowledgements

The authors are grateful to Drs. Thomas Schwarz, Max Heiman and Matt Pecot for critical feedback on this manuscript. We are particularly grateful to Dr. Takakazu Yokokura from the Okinawa Institute of Science and Technology for sharing an unpublished antibody against *Drosophila* Syndapin. We thank Drs. Nancy Bonini, Lynn Cooley, Jan Pielage, Simon Wang, Nicholas Harden, Charles Krieger, Norbert Perrimon and Mani Ramaswami for generously providing published genetic and antibody reagents. We recognize the Developmental Studies Hybridoma Bank for maintaining a variety of important antibody reagents. We thank Drs. Jennifer Waters and Talley Lambert and the Nikon Imaging Center at Harvard Medical School for technical support in light microscopy and methods of quantitative analysis. We thank Elizabeth Benecchi and the Harvard Medical School Electron Microscopy Core for assistance. We also acknowledge the contributions of Israel Pichardo-Casas, Deborah Pano, Jonathan Kibel and Colby Parsons to a screen for CRISPR alleles for miR-316. This work was supported by NIH grants NS069695 (D.V.V.) and NS090994 (D.V.V., D.P.W. & L.C.G.).

## Author contributions

E.M.M. and D.V.V. conceived and supervised project; E.M.M., C.W., A.T., H.H. performed screen and analysis of NMJ phenotypes and expression patterns; S.A. and L.C.G. performed EJP recordings; T.A.F. and E.M.M. created the miR-SP collection; T.F.D. and D.P.W. assisted in bioinformatics analysis; E.M.M. and D.V.V. wrote manuscript, with editorial suggestions from L.C.G.

## Competing interests

The authors declare no competing interests.
