## [Peer Review File · Nature Communications]

Reviewers' Comments:

Reviewer #1:

Remarks to the Author:

In the present study, McNeill et al. systematically investigated the effects of ubiquitous overexpression of 131 miRNA-Sponge (miR-SP) transgenes on bouton number of the larval *Drosophila* neuromuscular junction (NMJ). The genetic screen identified 21 candidate lines with altered bouton number. Further validation of these candidates with mutants yielded 5 candidate miRNAs. Based on evolutionary conservation and predicted synaptic function of the predicted targets, the authors focused on the candidate miR-34.

miR-34 null mutant NMJs have fewer boutons (~40% of control) with increased anti-Bruchpilot fluorescence intensity and decreased anti-Discs-Large (DLG) and Syndapin intensities. EM analysis uncovered a strong decrease in the area of the subsynaptic reticulum (SSR). AP-evoked synaptic transmission was largely unchanged in miR-34 null mutants. While ubiquitous or muscle-specific miR-34SP expression decreased bouton number, the opposite phenotype was observed after motoneuron-specific expression.

Considering miRNA sequence homology and a predicted synaptic function of the predicted miR-34 targets, they further analyzed *Nrx-IV* and *Adducin/Hts*. Using immunohistochemistry, they detected increased synaptic *Nrx-IV* and *Hts-M* levels in miR-34 null mutants. Moreover, motoneuron-specific *Nrx-IV* overexpression increased bouton number, and motoneuron-specific co-expression of miR-34SP and *nrx-IV-RNAi* rescued the increase in bouton number seen after presynaptic miR-34SP expression alone. Muscle-specific overexpression of *hts-M* decreased bouton number, while muscle-specific co-expression of miR-34SP and *hts-RNAi* yielded control bouton numbers. Finally, they noted a slight decrease in the number of nascent boutons upon KCl application in miR-34 null mutants or after muscle-specific *hts* overexpression.

Together, they conclude that presynaptic miR-34 negatively regulates *Nrx-IV*, thereby limiting *Brp* levels and bouton number. Postsynaptic miR-34 promotes synaptic growth by decreasing *Hts-M* levels.

Based on a miR-SP collection previously published by the same group (Fulga et al, 2015), the major strength of the paper is the systematic investigation of the role of miRs in synaptic development. The topic is timely and of interest to a general readership. The major limitations, which are detailed below, are (1) the limited evidence for links between miR-34 and *Nrx-IV* or *Hts/Adducin*, (2) no data on the relationship between *Nrx-IV* and *Hts-M* in producing the miR-34 phenotype, and (3) unclear statistics. Significant revisions and additional experiments are required to support the major conclusions. The following points (in chronological order) need to be addressed:

MAJOR POINTS

1) Except for figures 2f, S2-4, sample sizes are not reported throughout the manuscript. It is therefore impossible to judge the statistical power and the meaning of most p-values.

Regarding the genetic screen: How do sample sizes compare between controls and miR lines (Figure 1, Table S1)? Was the control continuously quantified during the screen? How meaningful are the statistics/ what is the statistical power of the p-values given in Table S1? Sample sizes should be reported for all data.

2) The average control bouton counts range between <120 (Figure S4d) and >140 (Figure 5f), and thus by ~20%. Some phenotypes are in the range of +/-20% (Figures 1b, 3e, 4e, 5e, 5f). Were the experimental groups of a given experiment collected side by side? Please report the respective effect sizes and comment on the variability of the control data with respect to the experimental groups.

1) The authors focus on miR-34 because of in silico ranking of the predicted miR targets with regard to predicted synaptic functions (SynaptomeDB) and homology (p. 5, 6; Figure S1). This seems

somewhat arbitrary, as other miRs have a very similar score. For instance, miR-92b has at most 2% fewer predicted targets with synaptic function, and only a few percent fewer conserved targets with synaptic function. Are these differences significant? Does the ranking consider the score/mean probability of SynptomeDB (Obregon et al., 2015) or algorithms predicting homology? Given the absolute number of targets, by how many genes do the miRs differ from one another?

2) Can the reduced bouton number/increased Brp intensity and reduced Dlg intensity/SSR area in miR-34 null mutants (Figure 2) be rescued by ubiquitous miR-34 expression?

3) As described in figure legend 2, Brp intensity was quantified from 3D-SIM data (Figure 2c). How exactly was Brp intensity quantified? How was the “computationally reconstructed” data masked? Where volumes analyzed? How does the SIM-based estimate compare to confocal data? Was the SIM data normalized to confocal HRP data? How variable was the HRP intensity? A lot more details are required to evaluate the subtle change in Brp intensity.

4) Brp intensity (Figure 2c) and Brp puncta number (Figure S2c) are increased in miR-34 mutants. At the same time, the synapse is significantly smaller (-40% boutons) and EM analysis revealed “abnormal and diffuse” AZ morphology (Figure 2g, h, S2g-j). The examples in figure S2g-j actually suggest smaller T-bars in miR-34 mutants. How does this compare to increased Brp intensity? Could T-bar width and/or Brp-puncta area be quantified? How does an increase in Brp intensity and puncta number go together with a smaller synapse? Representative Brp SIM images would be also very helpful.

5) Brp intensity (Figure 2c), Nrx-IV intensity (Figure 4c) and Hts-M intensity (Figure 5c) are increased in miR-34 null mutants. Could this be due to reduced SSR area (Figure 2f)? For instance, muscle-specific perturbation of alpha/beta-spectrin impairs the SSR and leads to increased Brp levels per GluR field (Pielage et al., 2006). Can the SSR phenotype and the increased Brp and NrxIV levels be rescued by postsynaptic Hts-RNAi expression?

6) miR-34 mutants do not have a defect in AP-induced EJP amplitude (Figure 2i-k). As miR-34 mutants have significant postsynaptic morphological defects – are there any changes in miniature EJP amplitude or muscle input resistance? (mEJP recordings and analysis are even described in the methods).

7) The increase in bouton number after presynaptic miR-34SP expression is confusing (Figure 3). Can the same result be obtained after postsynaptic rescue, i.e. expression of a miR-34 transgene in the miR-34 null mutant background?

8) Why is postsynaptic miR-34 epistatic with regard to bouton number? Is this also true for other aspects of the miR-34 phenotype? Does postsynaptic hts-RNAi expression rescue bouton number in the miR-34 null mutant background?

9) On page 9, they write: “MRE search algorithms could predict a large number of candidate targets (376), we knew that most of these would be irrelevant or non-functional at the synapse.” Which data support this statement?

10) Figures 4c and 5c: Similar to point 5), a lot more details are required with regard to Nrx-IV and HTS-M image analysis. Was presynaptic and/or postsynaptic fluorescence analyzed? How was the data masked? Why was the data normalized to the “area at synaptic arbor” (Figure 4c) and the “area of muscle 6/7 segment A2” (Figure 5c)? It would helpful to plot individual data points (average per NMJ) for the Nrx-IV/Hts-M and the HRP channel.

11) Two important controls are missing for the experiments shown in figures 4f and 5f: The effect of concomitant miR-34SP and nrxIV RNAi/Hts RNAi expression could be simply due to reduced/increased bouton number after nrxIV RNAi/Hts RNAi expression alone, respectively. Bouton number after nrxIV RNAi expression alone (Figure 4f) and hts RNAi expression alone (Figure 5f) should be therefore assessed.

12) Have nrxIV-RNAi and hts-RNAi been validated?

13) Could the effects shown in figures 4f and 5f be due to limited Gal4 expression/i.e. reduced miR-34SP expression?

14) Page 11: There was also a marked increase in hts-M expression throughout the muscle fibers

(compare Fig. 5a to 5b). These data should be quantified.

15) Similarly, the data underlying the statement "(...) Using a Dmef2-GAL4 driver, we found that a single transgene insertion (UAS-hts-MwtVK33) produced an elevation of Hts-M only slightly greater than the Hts-M elevation we observed in miR-34 mutants (Figure S4b, c)" (page 12) should be quantified.

16) What is the sample size for the data shown in figure 5g? Are 2-3 fewer nascent boutons (out of how many nascent boutons?) relevant (in the context of 120 boutons)?

MINOR POINTS

1) Supplementary text/legend for tables S1 and S2 are missing and should be added.

2) The graphs showing "Difference from control" (Figures 1b, 3e, 4e, 5e, and 5g) should also include SD/SEM of the respective controls.

3) The font size of the Y-axis of figures 2i-k is too small and should be increased.

4) Scale bars are missing in figure 4 (mentioned in legend).

5) Does the absolute data shown in Figure 4f correspond to the normalized data shown in figure 4e? The legend should specify if the same or different data are shown.

6) The labels "nrxIV LOF" (Figure 4f) and "hts LOF" (Figure 5f) should be changed to "nrxIV/hts RNAi".

7) The model does not capture that postsynaptic Hts-M drives nascent bouton formation.

8) Figure S2f: The TEM w1118 data set is based on n=3 larvae. Are the statistics based on n=3 larvae, presynaptic area, or "bouton image"?

Reviewer #2:

Remarks to the Author:

This is an important study reporting on the role of a microRNA, miR-34, in neuromuscular synapse formation in *Drosophila*. By an unbiased and systematic screening of loss-of-function of miRNAs, the authors identified an unexpectedly large number of miRNAs regulating NMJ development. The authors then performed cell-type specific and more detailed analyses of one of the identified microRNA, miR-34, and showed that miR-34 plays distinct roles in pre- and post-synaptic development by regulating the expression level of the junctional receptor Neurexin-IV and cytoskeletal effector Hts in motor neurons and muscles, respectively. Very interestingly, down-regulation of miR-34 in motor neurons and muscle showed opposite phenotypes on NMJ synapse formation. This study thus clearly demonstrates distinct and counteractive roles of microRNA in interacting cells. These findings would be of great interest to a wide range of researchers, who study regulation of developmental processes in general, and synapse formation in particular, and thus merit publication in *Nature Communications*. I only have a few comments that the authors may consider before publication of the manuscript.

Major points:

1. The strategy the authors used to prioritize their deeper studies on miR-34 among the ten verified genes identified in the initial screening is not clear to this reviewer. The authors first "compiled a database of 470 genes with known larval NMJ phenotypes and combined this with a list of dme genes orthologous to hsa genes with synaptic annotations (Supplemental Table 2). What are the genes listed in Supplemental Table 2? Genes that show larval NMJ phenotypes AND are orthologous to human synapse-regulating genes? Or, combination (OR) of the two? If the latter is the case, the authors should indicate which of the two categories the genes belong to. The authors then "ranked their novel miRNA candidates with these two filters as a function of enrichment for predicted synaptic target genes." What do the "two filters" refer to? A more detailed description of the selection process would be helpful.

2. Similarly, it is not clear how the authors narrowed down the miR-34 targets to just three genes out of the 56 candidate synaptic genes. How valid is the use of the "stringent criterion"? While the following experimental analyses showed that Neurexin-IV and Hts are the major downstream players, possible involvement of other candidate genes should not have been excluded at this point. Inclusion of a list of the 56 synaptic genes (and possibly some description of outstanding candidates in the text) would help the readers to comprehend a wider landscape of the downstream candidates.

3. Fig. 4f and Fig. 5f: these are key experiments showing the requirement of Neurexin-IV and Hts in causing the miR-34 phenotypes. However, one concern is that multiple UAS insertions might compete for a limited amount of Gal4. Thus, the rescue of the miR-34 phenotypes could be due to decreased expression of miR-34 SP. An appropriate control for these experiments would be miR-34 2xSP+control UAS insertion.

4. Fig. 4e: it is important to show that the level of NrXIV expression in NrXIV OE is comparable to that in miR-34 2XSP. Similarly, the level of Hts expression in Hts-M OE (Fig. 5e) should be quantified.

Minor points

1. As mentioned above, Supplemental Tables lack titles and legends, and are hard to interpret.

2. I wonder why just one target gene is the major target of miR-34 in each cell. A potential merit of microRNA regulation would be the ability to simultaneously control the expression of a number of downstream genes sharing the same MREs. The results were in contrast to this hypothesis. This may be a naïve question but readers might expect more illuminating discussion on this issue.

3. Figure 5a, b: difficult to see the increase in Hts intensity. Separate Hts expression (without co-labeling with HRP) should be presented.

4. Fig. 5e, g: statistical analyses between miR-34 and Hts-M OE should be included to show that the phenotypes are comparable.

Typos

Page 7, second paragraph: "The ultrastructure of normal 1b synaptic boutons is have been extensively characterized."

Page 11, second paragraph: "and is recognized the antibody"

Page 13, second paragraph: "we did not observed"

Page 13, second paragraph: "thus decreasing addition of nascent boutons significantly"---increasing?

Referee #1:

We appreciated this referee's careful attention to detail, their intellectual curiosity, and their encouraging statement that our "topic is timely and of interest to a general readership"; we have made an effort to make the quantitative detail and rigor of our study more obvious to the reader, and thank reviewer #1 for those suggestions. We feel that we should clarify two details noted in this reviewer's general comments, before answering the individual points. First, we reported ten (not five) novel microRNAs with highly significant NMJ phenotypes that are validated by comparison to independent, viable deletion alleles. Second, the loss of miR-34 function results in a complete loss of activity-induced nascent bouton addition normally triggered by spaced potassium depolarization in L3 larvae. In the original Fig 5, we plotted the miR-34 null and Hts-M-OE on a percentage difference graph due to the fact that each genotype requires a different genetically-matched control; however, we realize that this makes the phenotype appear weak, so we have replotted the results in raw "ghost bouton" numbers to emphasize the fact that our controls match the standards published by the leading labs in the field, and that miR-34 nulls completely prevent this activity-dependent bouton addition. The comparable impact of miR-34 loss and Hts-M-OE in this assay suggests that the postsynaptic regulation of Hts acts to control bouton initiation as opposed to a later step, such as maturation-dependent stabilization that might depend on factors in the SSR. We thank this referee for bringing this to our attention and hope the change in Fig. 5h and the edited text helps clarify this important point.

MAJOR POINTS (reviewer comments in red; renumbered due to redundancy of 1 & 2)

1a) Except for figures 2f, S2-4, sample sizes are not reported throughout the manuscript. It is therefore impossible to judge the statistical power and the meaning of most p-values. Regarding the genetic screen: How do sample sizes compare between controls and miR lines (Figure 1, Table S1)? Was the control continuously quantified during the screen? How meaningful are the statistics/ what is the statistical power of the p-values given in Table S1? Sample sizes should be reported for all data.

For the primary screen of 132 genotypes, sample size was standardized at 10 hemisegments of M6/7 NMJs per genotype; the average total number of boutons counted in this sample size was 1,224 in controls. A statistical power analysis that we performed before the screen suggested that a 10-hemisegment sampling depth, with a bouton count and variance (std) typical of our prior published work, would allow us to assign significance at a power of 0.8 to effects sizes at or above 19% change from genetically matched controls at $p < 0.05$. For this reason, only genotypes with bouton counts that differ from matched controls at a significance at or exceeding $p < 0.05$ were taken to the next level of analysis with available null alleles. We have clarified this in the methods, but to help make the screen sample size more accessible to the reader, we have also added this information in the main text. With regards to controls, since the laborious histology had to be performed in subsets (10-12 genotypes plus controls per batch), a *scrambleSP* control was included continuously in each subset prepared; we now clarify this in the methods.

Setting the sensitivity threshold of our primary screen at this level was largely a practical decision given the laborious nature of the assay, but this choice certainly limited our ability to detect subtle phenotypes. Moreover, as we point out in the text, viable deletion alleles are not available for all of our hits. Thus, our screen probably represents an underestimate of the novel microRNAs that regulate NMJ morphogenesis.

For deeper follow up phenotypic analysis after the primary screen (e.g. miR-34 targets and rescue etc.), our standard sample size was roughly n=20 hemisegments per genotype. We have checked to be sure that sample size is described in all of our figure legends. Importantly, every follow up experiment in the study included controls that were genetically matched as closely as possible to the experimental genotype(s). This is precisely why we display our data as percentage change when comparing phenotypes that require different control backgrounds (e.g. *scrambleSP* versus *miR-SPong*e compared to miR deletion and its corresponding wild type background control). We have added notes to clarify this in the methods.

2a) The average control bouton counts range between <120 (Figure S4d) and >140 (Figure 5f), and thus by ~20%. Some phenotypes are in the range of +/-20% (Figures 1b, 3e, 4e, 5e, 5f). Were the experimental groups of a given experiment collected side by side? Please report the respective effect sizes and comment on the variability of the control data with respect to the experimental groups.

In our study, the control backgrounds are different for different experiments and are always collected side-by-side. In our field it is well known that control genotypes need to be carefully matched to experimental genotypes, as certain backgrounds such as *w¹¹¹⁸* or *GAL4* drivers have an impact on NMJ size compared to classical wild type backgrounds such as Canton-S. This is the reason for the range of control values. This is also a benefit of using a collection like miR-SP where all lines and control (scramble) are made in the same genetic (AttP) background. In all cases reported in Fig. 1b, the difference between *GAL4;miR-SP* and *GAL4;scrambleSP*, or the difference between *DfmiR* and the background control provided by the Cohen group, meet or exceed a significance threshold of $p \leq 0.05$. The effect sizes in percentage change for novel *miR-SP* phenotypes reported for the primary screen in Fig.1 ranged from 19% to 42%. Indeed, the robust strength and consistency of the miR-34 bouton phenotype was one of the factors leading to our decision to pursue deeper analysis. We have added a comment in the text to clarify this.

1b) The authors focus on miR-34 because of in silico ranking of the predicted miR targets with regard to predicted synaptic functions (SynaptomeDB) and homology (p. 5, 6; Figure S1). This seems somewhat arbitrary, as other miRs have a very similar score. For instance, miR-92b has at most 2% fewer predicted targets with synaptic function, and only a few percent fewer conserved targets with synaptic function. Are these differences significant? Does the ranking consider the score/mean probability of SynaptomeDB (Obregon et al., 2015) or algorithms predicting homology? Given the absolute number of targets, by how many genes do the miRs differ from one another?

Although our prioritization for deeper analysis was based in part on the number of predicted targets known to have some synaptic function in fly and vertebrates, this referee is correct that we also applied additional criteria. In the case of miR-34, in addition to the robust phenotype,

our discovery of distinct effects in presynaptic and postsynaptic cells offered an intriguing opportunity to test the cell type specificity of downstream targets. One key criterion for deeper analysis of targets was the degree to which a predicted target gene was conserved from fly to human, both for protein coding sequence AND for MRE seed sequence complementarity (also see Rev 2 point 2). Thus, the ranking in Suppl. Fig. 1b does incorporate homology prediction (using Diopt to identify *Drosophila* homologs of genes in SynaptomeDB). In the case of the miR-92 family, the confounding issue was the existence of two close orthologs (miR-92a and miR-92b) and several additional insect family members (miR-310-313) that share overlapping seed sequences. Moreover, miR-92a and miR-92b are both embedded in a shared host gene (*Jigr*) that itself is required for early stages of neuronal stem cell development (as shown by our colleagues in the Gao lab). This made the genetics and phenotypic analysis of miR-92 both complex and difficult to dissect in the short term. However, we hope to explore this and other microRNA families in the long term. We have modified the text to reflect this.

2b) Can the reduced bouton number/increased Brp intensity and reduced Dlg intensity/SSR area in miR-34 null mutants (Figure 2) be rescued by ubiquitous miR-34 expression?

Unfortunately, as shown in a previous paper that we contributed to from the labs of Drs. Eric Lai and Norbert Perrimon, broad over-expression of microRNAs (*in vivo* mimics) causes lethality in a very high percentage of cases (Bejarano et al., 2012), even when complete nulls are viable (Chan et al., 2014); miR-34 is no exception. For example, UAS-miR-34(+) displays lethality with *da-Gal4* (Bejarano et al., 2012) even though the null is viable. We did find that more selective elevation of miR-34 in motor neurons with *OK6-Gal4* allowed animals to survive to L3 (although they die shortly afterwards in pupation). Therefore, we counted type 1 boutons at the 6/7NMJ and found that increased presynaptic miR-34 has the opposite effect of the miR-34SP in motor neurons, thus supporting the model where presynaptic miR-34 acts to promote NMJ growth. We added this new finding to Fig. 3e, as the results were highly significant. In parallel, we attempted to use UAS-miR-34(+) to elevate microRNA levels selectively in muscle, however this caused embryonic and very early larval lethality, thus preventing postsynaptic analysis or rescue of the late larval NMJ.

3) As described in figure legend 2, Brp intensity was quantified from 3D-SIM data (Figure 2c). How exactly was Brp intensity quantified? How was the “computationally reconstructed” data masked? Where volumes analyzed? How does the SIM-based estimate compare to confocal data? Was the SIM data normalized to confocal HRP data? How variable was the HRP intensity? A lot more details are required to evaluate the subtle change in Brp intensity.

Intensity and volume were quantified using different tools in ImageJ. For Brp intensity profiling, we initially analyzed laser scanning confocal data (acquired on a Nikon A1R) because we wanted to replicate the methods used by the investigators that first described the effect of *Nrx-IV* OE on the active zone marker. We did observe intensity changes using confocal. However, when we moved to volume measurements and object counting, we transitioned to 3-D SIM (OMX Blaze) for all subsequent measurements because AZ morphology (including the central region of low Brp density in each AZ) was poorly resolved on our confocal.

Brp was measured by 3D-SIM in samples co-stained with neuron-specific anti-HRP, and the synaptic mask within the terminal boutons of an arbor branch was generated with a combination of ImageJ tools (e.g. **Analyze Particles** tool) that are described in much greater detail in the revised methods section. The “computational reconstruction” was simply 3-D volume rendering from the Z-stack, which was necessary for volume measurements of the Brp signal using the **3-D Objects Counter** tool; this is now described in greater detail in the methods section. It is important to note that the synaptic biomarkers Brp and NrX-IV analyzed with 3-D SIM were captured from the terminal boutons in each arbor branch, thus avoiding bias introduced by the number of boutons per NMJ. Finally, HRP signal did not show significant differences between *miR-34* null and control ($p=0.33$; see graph inset on right) and had variance comparable to most other markers; however, it is important to note that HRP signal intensity was not used to normalize the Brp or other intensity signals. The normalization that we performed for Brp or NrX-IV was as a function of area captured by the synaptic mask that was created using the HRP membrane outline, so that we could see the fold-change in the experimental value compared to control (defined as a value of 1.0). This approach corrects for the area (or bouton size) captured. We apologize for confusing the reader with a lack of sufficient detail in the methods.

4) Brp intensity (Figure 2c) and Brp puncta number (Figure S2c) are increased in *miR-34* mutants. At the same time, the synapse is significantly smaller (-40% boutons) and EM analysis revealed “abnormal and diffuse” AZ morphology (Figure 2g, h, S2g-j). The examples in figure S2g-j actually suggest smaller T-bars in *miR-34* mutants. How does this compare to increased Brp intensity? Could T-bar width and/or Brp-puncta area be quantified? How does an increase in Brp intensity and puncta number go together with a smaller synapse? Representative Brp SIM images would be also very helpful.

Unlike normal T-bars, the *miR-34* mutant T-bars are very difficult to measure in TEM sections precisely because of their somewhat diffuse and irregular appearance. We did quantify Brp puncta in the original Supplementary Fig. 2, from 3-D SIM IHC imaging data, and we consistently see a two-fold increase in the number of objects. These Brp clusters often appear larger than normal; we added 3-D SIM images to Supplementary Fig. 2 (a-f) to help the reader see what the light level structures look like in the OMX Blaze. However, we cannot conclude that all Brp clusters in these mutants form functional T-bars that would be recognizable in the TEM. Unfortunately, we have not yet mastered correlative light and EM microscopy for structures like T-bars. Interestingly, the mEJP frequency increase does suggest that the increased Brp is correlated with higher spontaneous release events, which might reflect some abnormal stoichiometry in the *miR-34* null active zones. Perhaps this two-fold increase in the number of puncta also helps account for the relatively normal EJP output despite a nearly 2-fold decrease in boutons. Of course, this is only speculative.

5) Brp intensity (Figure 2c), NrX-IV intensity (Figure 4c) and Hts-M intensity (Figure 5c) are increased in *miR-34* null mutants. Could this be due to reduced SSR area (Figure 2f)? For

instance, muscle-specific perturbation of alpha/beta-spectrin impairs the SSR and leads to increased Brp levels per GluR field (Pielage et al., 2006). Can the SSR phenotype and the increased Brp and NrXIV levels be rescued by postsynaptic Hts-RNAi expression?

If this is a technical question about the way intensities were measured in our study, then, we believe not. In our initial manuscript, we used two types of masks for quantification of intensity: **(1)** A synaptic mask that was based on HRP in high magnification images to define the perimeter of the terminal boutons as a “region of interest” (ROI); this was used for Brp and NrX-IV. Or **(2)** a much larger ROI defined in lower magnification images of the entire NMJ by a rectangular box that fits around each whole NMJ and extends well beyond the SSR boundary in all cases; this was used specifically for Hts-M IHC intensity measurements. Because neither of these masks were based on measured SSR dimensions, a phenotypic change in SSR area would not significantly alter the ROI dimensions in any of the genotypes measured. Had we defined the ROI with Syndapin, that would potentially have been an issue, but we did not.

However, if this is a biological question regarding the epistatic relationship between SSR and miR-34 targets or other biomarkers, it is worth commenting. We believe it unlikely that Hts-M intensity increases because of reductions in SSR, partly because SSR is the most obvious source of determinants for Hts-M localization in wild type NMJs (relatively little Hts-M is detected outside of the SSR in controls). Moreover, published work has shown that Dlg localization is dependent on Hts, and SSR requires Dlg. It thought that Hts-M/Adducin family proteins participate in a synaptic cytomatrix, which includes not only Spectrin/Ankryn networks, and F-actin, but also signaling and adhesion molecules. Thus, it may be possible that disrupting the stoichiometric balance of components in the protein network may lead to further changes in the cytomatrix. However, the impact of spectrin/ankryn function on SSR could just as easily be a downstream output of elevated Hts-M/adducin in the *miR-34* mutants. No one fully understands the sequence of events or protein activities that result in SSR formation, maintenance or distortion.

6) miR-34 mutants do not have a defect in AP-induced EJP amplitude (Figure 2i-k). As miR-34 mutants have significant postsynaptic morphological defects – are there any changes in miniature EJP amplitude or muscle input resistance? (mEJP recordings and analysis are even described in the methods).

The mean input resistance recorded for control was **7.93 ± 0.16** megaohms (n=7) versus **8.24 + 0.19** megaohms (n=8) for *miR-34* null (p=0.4178; a non-significant difference as assessed by Kruskal-Wallis test), and all samples tested had to display a stable resting membrane potential for the duration of the recording. Interestingly, while there was no significant difference in the mean amplitude of miniature EJP events in *miR-34* mutants, we did see a significant elevation in mEJP frequency. This may reflect changes in the composition of active zones suggested by the observations described in point 4 above, but we felt that this might be too speculative to add to the discussion. The mEJP data are now shown in Fig. 2, and the input resistance data are included in the legend.

7) The increase in bouton number after presynaptic miR-34SP expression is confusing (Figure 3). Can the same result be obtained after postsynaptic rescue, i.e. expression of a miR-34 transgene in the miR-34 null mutant background?

We agree that the distinct impact of miR-34 inhibition in motor neurons versus muscle tissue is intriguing and complex; we believe that this reflects the contributions of multiple target genes and cellular sites of action. As outlined in item 2b above, the *UAS-miR-34(+)* transgene was early lethal when expressed in muscle, and thus could not provide an opportunity to perform the suggested rescue experiment. However, elevating miR-34 levels with motor neuron overexpression of *UAS-miR-34(+)* resulted in a highly significant reduction in type 1 bouton number from **146.4** in *w¹¹¹⁸;OK6-GAL4* (n=20 A2 6/7NMJ) to **103.2** in *w¹¹¹⁸;OK6-GAL4;UAS-miR-34(+)* (n=20 A2 6/7NMJ) [$p \leq 7.5 \times 10^{-5}$]; this is precisely the opposite of the motor neuron *miR-34SP* inhibition. Thus, this new finding (shown in Fig. 3e) now helps reinforce the model that presynaptic miR-34 activity can modulate presynaptic growth, even though we believe that budding of new nascent boutons at terminal type 1 bouton cannot escape the postsynaptic restriction imposed by miR-34 and Hts in muscle. As stated elsewhere, we currently think that this reflects a requirement for coordinated morphogenetic (and possibly adhesive) plasticity on both sides of these synaptic structures to enable formation of new structures.

8) Why is postsynaptic miR-34 epistatic with regard to bouton number? Is this also true for other aspects of the miR-34 phenotype? Does postsynaptic hts-RNAi expression rescue bouton number in the miR-34 null mutant background?

At this referee's suggestion, we combined the *miR-34* null mutant with an independent muscle-specific driver (*Mhc-GAL4*) to express *UAS-Hts-RNAi* in order to test the importance of Hts regulation in causing the *miR-34* NMJ phenotype. Postsynaptic Hts knock-down is indeed sufficient to significantly rescue the NMJ bouton number defect of the *miR-34* null from **92.42** type 1 boutons per A2 6/7 NMJ (n=19) in *Mhc-GAL4;miR34⁻/miR34⁻* to **141.05** (n=20) in the *Mhc-GAL4;UAS-hts-RNAi;miR34⁻/miR34⁻* [$p \leq 0.001$]. This is now shown in Fig 5g. We are very grateful for this suggestion, as it provides even stronger evidence that Hts is required for the major miR-34 NMJ growth defect.

At type 1 boutons, nascent buds form within the space occupied by SSR and many adhesive contacts between motor neuron and muscle, so it is not that surprising that SSR needs to be altered in some way to accommodate new terminal growth. Although we did avoid the term epistatic, our internal working model posits that postsynaptic structural remodeling is required in tandem with coordinated presynaptic morphogenesis, and that interference at the level of cytoskeletal protein effectors can disrupt SSR remodeling downstream of signaling pathways that would normally manage the coordination. This is obviously a bit speculative to include in our current manuscript, and will require many future experiments to identify the relevant anterograde signaling pathway(s). We have tried to streamline the text and edit Figure 5i, to clarify a basic working model that miR-34 and Hts control SSR remodeling to allow presynaptic budding/expansion.

9) On page 9, they write: "MRE search algorithms could predict a large number of candidate targets (376), we knew that most of these would be irrelevant or non-functional at the synapse." Which data support this statement?

The cited review (Riffo-Campos et al., 2016) covers papers that support the statement that for most microRNAs studied to date, very few of the predicted targets have been shown to exhibit relevant biological activity. However, we appreciate this referee's concern over the implication

and tone of the sentence, and thus we have edited the sentence to present a more conservative point of view.

10) Figures 4c and 5c: Similar to point 5), a lot more details are required with regard to NrX-IV and HTS-M image analysis. Was presynaptic and/or postsynaptic fluorescence analyzed? How was the data masked? Why was the data normalized to the “area at synaptic arbor” (Figure 4c) and the “area of muscle 6/7 segment A2” (Figure 5c)? It would be helpful to plot individual data points (average per NMJ) for the NrX-IV/Hts-M and the HRP channel.

We apologize for the insufficient information and agree that more methodological detail would be helpful for the reader in both understanding and replicating our results, so we have added text on our analysis using ImageJ tools to the methods sections. Two types of masks were used for different data. The method of masking for presynaptic NrX-IV intensity profiling was identical to the approach used for Brp (discussed in point 5) and focused on the terminal type 1 boutons using 3-D SIM data (discussed above in point 3). However, because Hts-M concentrates in a large postsynaptic corona surrounding the motor neuron and is also elevated across the muscle fiber as well, Hts-M intensity profiling was performed with lower magnification images captured on a laser scanning confocal. Both analyses required an ROI to be defined, however, the ROI for Brp and NrX-IV in 3-D SIM image stacks was defined using a presynaptic marker to define the perimeter of the terminal. In the case of Hts-M, we used an ROI surrounding the entire NMJ that could accommodate all the images in a dataset for consistency. Thus, the ROIs used to measure Hts-M were selected by us and always extended beyond the presynaptic terminal itself and incorporate some Hts-M signal in the muscle field. We standardized this ROI for each dataset so that it could be applied to control and mutant in the same way for consistency. Below is a set of data comparing control and miR-34 null which illustrates that Hts-M antigen is not significantly elevated when we apply an HRP-defined synaptic mask (first graph); however when we measure Hts with a mask that includes the whole NMJ and some of the surround (second graph; this method is used for Fig. 5c), or if we only image the surrounding muscle using a set of 5 small ROIs outside of the NMJ (third graph with green bar), we see a significant increase.

The normalization in each case, as in the case of synaptic markers like Brp or NrX-IV, is to the area of the ROI so that we can see the fold change relative to control set at a value of 1.0.

11) Two important controls are missing for the experiments shown in figures 4f and 5f: The effect of concomitant miR-34SP and nrxIV RNAi/Hts RNAi expression could be simply due to reduced/increased bouton number after nrxIV RNAi/hts RNAi expression alone, respectively. Bouton number after nrxIV RNAi expression alone (Figure 4f) and hts RNAi expression alone (Figure 5f) should be therefore assessed.

We agree that the reader needs to know the impact of cell type specific knock down for NrxF and Hts alone; these experiments are described in the text and have been added to Supplemental Figures 3 (*OK6-GAL4;UAS-nrx-IV^{RNAi}*) and 4 (*DMef2-GAL4; UAS-hts^{RNAi}*). In the case of Hts knock down in muscle, there was no significant impact on bouton number with this driver/RNAi combination. However, motor neuron-specific knock down of NrxF, which has not been described in previous papers, led to a significant increase in bouton number. This could not have compensated for miR-34 inhibition in a simple additive way in the observed rescue, thus ruling out the caveat mentioned above. However, we were surprised to see this *UAS-Nrx-IV^{RNAi}* phenotype, which is very similar to presynaptic over-expression. One possible explanation for this may be that there is an optimum level of NrxF, and expression above or below this optimum interferes with the stability of the presynaptic varicosity, allowing added growth. Such biphasic functional relationships and optima are characteristic of some adhesion molecules, but this is not known for neuronal NrxF.

12) Have nrxF-RNAi and hts-RNAi been validated?

Yes, the *UAS-RNAi* lines for both NrxF and Hts were validated by the authors that published the corresponding papers which we cited in the text. For this reason, we only used published lines that were available from the Bloomington stock center or by request from the previous authors.

13) Could the effects shown in figures 4f and 5f be due to limited Gal4 expression/i.e. reduced miR-34SP expression?

As requested explicitly by reviewer #2, we performed additional control experiments to determine if adding a second UAS construct in the *miR-SP* background might dilute and attenuate Gal4 activity, and thus weaken the effect of the SP in our rescue experiments. We now show in Supplemental Figures 3 (for *OK6-GAL4*) and 4 (for *DMef2-GAL4*) that adding one of two different *UAS-GFP* reporters, instead of *UAS-RNAi*, has no significant impact on the phenotypic penetrance of either the scramble-SP control or the *UAS-miR-34SP*. We thank both reviewers for asking this question because the control provides an important validation of the cell type-specific rescue. In addition, we performed a rescue experiment where the miR-34 null was combined with *UAS-hts^{RNAi}* expressed in muscle, and we see highly significant rescue of the NMJ growth phenotype (described above). This also provides strong support our working model that postsynaptic regulation of Hts plays an important role in the under-growth phenotype of miR-34 mutants.

14) Page 11: There was also a marked increase in hts-M expression throughout the muscle fibers (compare Fig. 5a to 5b). These data should be quantified.

As we outlined above and in the expanded methods section, a significant amount of muscle fiber area is already included in the ROI used for Hts-M quantification. Moreover, as shown in the graphs above (point 10), the intensity change observed in our Hts-M capture ROI is nearly the same as the upregulation in muscle outside of the NMJ.

15) Similarly, the data underlying the statement “(...) Using a *Dmef2-GAL4* driver, we found that a single transgene insertion (*UAS-hts-MwtVK33*) produced an elevation of Hts-M only slightly greater than the Hts-M elevation we observed in *miR-34* mutants (Figure S4b, c)” (page 12) should be quantified.

As also explained for Rev2 point 4, we ran out of the small aliquot of anti-Hts-M peptide antibody from Lynn Cooley’s lab, and they didn’t have additional aliquots that they could send us for the revisions. Thus, we had an affinity-purified antibody made to precisely the same synthetic peptide, but this took time. Using this new Rabbit anti-Hts-M peptide antibody (described in the Methods section), we added a quantitative analysis of Hts-M staining intensity comparing *DMef2-GAL4* crossed to *w¹¹¹⁸* as control, to *DMef2-GAL4* driving *UAS-hts^{RNAi}* (red bar; highly significant reduction), *UAS-hts-M^{VK33}* (1x) and *UAS-hts-M* (2x) from the Pielage lab (green bars; highly significant increase) to show the relative strength of the single and double insert (now in Supplementary Fig. 4). I apologize that my staff misunderstood me, and only imaged/quantified *miR-34* null heterozygotes over *DMef2-GAL4*, which do not show a significant bouton phenotype, instead of the homozygous null. Thus, we don’t have the direct comparison of null versus 1x*UAS-htsOE* yet, but a new experiment will soon be underway to remedy that. Although I want to see this result myself, I feel that I should stress that the key observation is that of genetic phenocopy and rescue. With the transgenics at hand, it is unrealistic to expect a perfect match; this is why we evaluated the 1x and 2x lines “by eye” to be sure that they levels were not wildly different.

16) What is the sample size for the data shown in figure 5g? Are 2-3 fewer nascent boutons (out of how many nascent boutons?) relevant (in the context of 120 boutons)?

As shown in papers from the Budnik lab that pioneered this assay, and others that have used this effectively (cited in our text), in control genotypes, 5 spaced KCl bath exposures to L3 larval pelts induces a highly significant ($p \leq 0.001$) increase in the formation of nascent boutons per NMJ relative to mock-treated controls (green bars in Fig. 5h). Each L3 A2 M6/7 NMJ in untreated or mock-treated control samples contains 0-3 nascent “ghost” boutons, whereas spaced depolarization pushes this up to 4-6. Our sample size for these spaced depolarization experiments varied from 11-24, and the specific values are described in the legend of Fig 5.

MINOR POINTS

1) Supplementary text/legend for tables S1 and S2 are missing and should be added.

Titles and legends have been added for Supplementary Tables 1 & 2.

2) The graphs showing “Difference from control” (Figures 1b, 3e, 4e, 5e, and 5g) should also include SD/SEM of the respective controls.

SEM values for each respective control have been added to the figure legends.

3) The font size of the Y-axis of figures 2i-k is too small and should be increased.

The font size has been increased and changed to bold for improved visibility.

4) Scale bars are missing in figure 4 (mentioned in legend).

A scale bar has added for Fig. 4a; the pixel dimension for the 3-D SIM was 40 nm.

5) Does the absolute data shown in Figure 4f correspond to the normalized data shown in figure 4e? The legend should specify if the same or different data are shown.

No. The overexpression and rescue were performed with independent, background matched controls. The *OK6-GAL4;ScrambleSP* genotype is the same in the two experiments, but the controls were each performed en batch with the separate experiments on different occasions. We note this in the figure legend.

6) The labels “nrxIV LOF” (Figure 4f) and “hts LOF” (Figure 5f) should be changed to “nrxIV/hts RNAi”.

The graphs have been updated with “RNAi” for clarity.

7) The model does not capture that postsynaptic Hts-M drives nascent bouton formation.

We have modified the cartoon and legend of Fig. 5 in order to clarify the working model.

8) Figure S2f: The TEM w1118 data set is based on n=3 larvae. Are the statistics based on n=3 larvae, presynaptic area, or “bouton image”?

The sample size for comparisons of AZ parameters in TEM data were n=12 boutons for control and n=16 boutons for *miR-34* nulls; this is described in the Fig. 2 legend.

Referee #2.

We greatly appreciate this referee’s thoughtful questions, and their opinion that this is an “important study” that “clearly demonstrates distinct and counteractive roles of microRNAs in interacting cells” – as this was our principal message and conclusion. We also appreciate the added opinion that our findings “would be of great interest to a wide range of researchers...and thus merit publication in Nature Communications.”

Major points (reviewer comments in red)

1. The strategy the authors used to prioritize their deeper studies on miR-34 among the ten verified genes identified in the initial screening is not clear to this reviewer. The authors first “compiled a database of 470 genes with known larval NMJ phenotypes and combined this with a list of *dme* genes orthologous to *hsa* genes with synaptic annotations (Supplemental Table 2). What are the genes listed in Supplemental Table 2? Genes that show larval NMJ phenotypes AND are orthologous to human synapse-regulating genes? Or, combination (OR) of the two? If the latter is the case, the authors should indicate which of the two categories the genes belong to. The authors then “ranked their novel miRNA candidates with these two filters as a function

of enrichment for predicted synaptic target genes.” What do the “two filters” refer to? A more detailed description of the selection process would be helpful.

As discussed in response to referee #1, we prioritized miR-34 for multiple reasons, including its robust phenotype, its distinct effects in motor neurons and muscles, and its degree of conservation with mammalian miR-34-family miRs. However, target analysis was a significant part of the rationale. Our prioritization and target identification strategy was designed for efficiency to take advantage of pre-existing data on synapse function in *Drosophila* and other models, as this is a strength of the NMJ in fly. To find potential targets that had already been implicated in synapse function, we combined two sets of genes (1: those previously identified by genetics in one of multiple prior *Drosophila* screens to have NMJ functions (470 total listed in Table S2); and 2: the Diopt-identified *Drosophila* orthologs of vertebrate genes annotated for synaptic function in SynptomeDB (just over 1000 total). The combination of these two lists of *Drosophila* genes was then used to search for predicted seed sequence (MRE) targeting by any of the ten novel microRNAs that hit in our screen using TargetscanFly. The number of these predicted targets were then used to rank each of the ten novel microRNAs that we validated with nulls (Supplementary Fig 1b). To clarify the process of prioritization, we edited the text in this section.

2. Similarly, it is not clear how the authors narrowed down the miR-34 targets to just three genes out of the 56 candidate synaptic genes. How valid is the use of the “stringent criterion”? While the following experimental analyses showed that Neurexin-IV and Hts are the major downstream players, possible involvement of other candidate genes should not have been excluded at this point. Inclusion of a list of the 56 synaptic genes (and possibly some description of outstanding candidates in the text) would help the readers to comprehend a wider landscape of the downstream candidates.

Because many microRNAs are known to acquire new species-specific targets at a rate higher than other regulatory factors (citation 54 in the text), and we wanted in general to examine conserved miR-to-target relationships, we added a second, more “stringent criterion” for prioritization of candidate targets before beginning functional analysis. Of the predicted targets from our database filter (described above), we asked which of these were sequence-conserved from fly to human at BOTH the predicted protein coding level AND the MRE level. We hoped this would enrich for targets that might be regulated by the corresponding microRNA(s) in both insects and vertebrates. We felt that this could be important to our prioritization because none of the previously characterized miR-34 targets that modulate synapse form and function display conserved miR-34-family MREs in *Drosophila*. We have edited the text to clarify this.

3. Fig. 4f and Fig. 5f: these are key experiments showing the requirement of Neurexin-IV and Hts in causing the miR-34 phenotypes. However, one concern is that multiple UAS insertions might compete for a limited amount of Gal4. Thus, the rescue of the miR-34 phenotypes could be due to decreased expression of miR-34 SP. An appropriate control for these experiments would be miR-34 2xSP+control UAS insertion.

Both of our reviewers point out this important potential caveat, which we have addressed in two ways as also described in our response to reviewer 1 (point 13 above):

- a) We showed that introduction of a second *UAS-GFP* transgene into the *UAS-miR-34SP* background does not significantly dilute either the *OK6-GAL4* or *DMef2-GAL4* drivers. The combined miR-34SP+UAS-GFP displays an NMJ phenotype equally penetrant to miR-34SP alone (now in Supplementary Figs. 3 and 4).
- b) We have combined the *miR-34* null allele with a muscle-specific RNAi knock-down of Hts. This reduction of Hts in the postsynaptic compartment is sufficient to restore NMJ bouton addition, thus rescuing the muscle function of miR-34 (now shown in Fig. 5g). This experiment also helps to confirm that postsynaptic regulation of Hts imposes a limit on bouton addition that cannot be overcome by loss of presynaptic miR-34.

4. Fig. 4e: it is important to show that the level of NrXIV expression in NrXIV OE is comparable to that in miR-34 2XSP. Similarly, the level of Hts expression in Hts-M OE (Fig. 5e) should be quantified.

During the period after submission of the original manuscript we ran into difficulties with antibodies. In the case of anti-Hts-M raised by Lynn Cooley's lab in Rabbit, they simply ran out of this and could not supply us with more. This led us to contract with a local company (Primm Lab) to make a new anti-peptide antibody in rabbits; which took time, of course. We recently began characterizing this new antibody (now described in the Methods), which shows the same SSR localization as the previous polyclonal. Using this new reagent, we added a panel to Supplemental Fig. 4 showing (with muscle-specific *DMef2-GAL4*) that, relative to a w^{1118} control, *UAS-hts^{RNAi}* does reduce the muscle signal, and that *UAS-hts-M^{VK33}* increases signal, but not to the high level expressed by the double insertion line *UAS-hts-M^{VK33,ZH51C}*. In the case of anti-Nrx-IV, the antibody that we used for this study was obtained from Dr. Christian Klambt at the University of Munster; however, we exhausted the initial aliquot and then requested another. He was kind enough to send a second aliquot recently, however, the second batch of antibody did not show any in situ staining in preps where the counterstains were all fine. Thus, we regret to say that we have not been able to perform additional experiments to quantify NrX-IV levels in the over-expressor strain in the short time since we found the new antibody stock to be useless; there has not been enough time to make a new antibody, or request different batches from Dr. Klambt.

Minor points

1. As mentioned above, Supplemental Tables lack titles and legends, and are hard to interpret. We have added titles and legends to the Supplementary Tables to clarify the data display.

2. I wonder why just one target gene is the major target of miR-34 in each cell. A potential merit of microRNA regulation would be the ability to simultaneously control the expression of a number of downstream genes sharing the same MREs. The results were in contrast to this hypothesis. This may be a naïve question but readers might expect more illuminating discussion on this issue.

As the reader suggests, it remains entirely possible that there are multiple targets which mediate the regulatory effects of miR-34 (or other microRNAs) in each cell type where it is active. Although we chose to focus on targets whose MRE regulatory elements suggest a conserved functional relationship with miR-34-family microRNAs from ecdysozoa to vertebrata,

and the rescue activity of these candidates was quite striking, we would not rule out contributions of other predicted targets even if these might be specific to insect species. We have modified our discussion to reflect the fact that this question is open for future investigation.

3. Figure 5a, b: difficult to see the increase in Hts intensity. Separate Hts expression (without co-labeling with HRP) should be presented.

We have modified the figure to show the separate channels with higher magnification that should allow the reader to see the local difference in Hts-M level and distribution.

4. Fig. 5e, g: statistical analyses between miR-34 and Hts-M OE should be included to show that the phenotypes are comparable.

The Fig 5e legend now includes a P-value of for a comparison between miR-34 and Hts-M OE ($p=0.79$).

Typos

Page 7, second paragraph: "The ultrastructure of normal 1b synaptic boutons is have been extensively characterized."

Page 11, second paragraph: "and is recognized the antibody"

Page 13, second paragraph: "we did not observed"

Page 13, second paragraph: "thus decreasing addition of nascent boutons significantly"---
increasing?

The old typos have been corrected or overwritten – thank you.

Reviewers' Comments:

Reviewer #1:

Remarks to the Author:

The authors have adequately addressed most of my concerns. In my view, the paper is now acceptable for publication in its current form.

Reviewer #2:

Remarks to the Author:

The authors properly addressed all of my major concerns. The manuscript can now be published in the journal.